# BAS: Bridging Adam and SignSGD for Memory-Efficient LLM Training

Yijie Zhou [1]   Mingliang Zhang [2]   Jiaqi Zhang [2]   Xunliang Cai [2]   Shi Pu [1]

## Abstract

We propose **Block Adaptive Signum (BAS)**, which bridges Adam and SignSGD via block-wise scaling of sign updates. By discarding element-wise second moments, BAS reduces memory overhead relative to AdamW without sacrificing performance in our tested settings. Crucially, BAS mimics Adam's dynamics closely enough to directly **inherit its hyperparameters**, matching the performance of AdamW without the need for re-tuning, a common fragility of prior low-memory optimizers. This structural alignment makes it particularly suitable for tuning Adam-pretrained models. Furthermore, we exploit the inherent robustness of sign-based updates to store the first moment in FP8 without performance degradation. This shrinks the optimizer-state footprint to **12.5% of AdamW's**. We theoretically prove convergence under standard assumptions and introduce a communication-efficient variant enabled by the sign-based update. Across extensive evaluations, including pre-training a 1.5B model on 100B tokens and supervised fine-tuning of models up to 32B parameters, we demonstrate that BAS achieves performance on par with AdamW.

## 1. Introduction

The rapid advancement of Large Language Models (LLMs) has revolutionized artificial intelligence, enabling capabilities that extend far beyond simple text generation, encompassing complex tasks such as multi-step reasoning, code generation, and autonomous agent planning (Zheng et al., 2025). However, training these models requires immense computational resources, particularly in terms of memory. Standard optimizers like Adam (Kingma & Ba, 2017) and

AdamW (Loshchilov & Hutter, 2019) incur significant memory overhead by maintaining high-precision first ($m$) and second ($v$) moments. Under modern mixed-precision training, where gradients are stored in BF16, these FP32 moment buffers introduce an optimizer-state overhead of $4\times$ the gradient memory. This memory overhead becomes a critical bottleneck in distributed training. High optimizer memory consumption limits the maximum local batch size, often forcing practitioners to scale out to more devices to fit the training state. This scaling introduces significant communication latency and infrastructure complexity, particularly when crossing inter-node interconnects.

To address this memory bottleneck, researchers have explored a range of memory-efficient alternatives. Early sign-based methods, such as SignSGD and Signum (Bernstein et al., 2018), showed that using only gradient or momentum signs greatly reduces optimizer state. However, these approaches struggled to match the optimization quality required for modern Transformer training. More recently, Lion (Chen et al., 2023) revived sign-based updates with improved empirical performance, though its reduced adaptivity introduces notable hyperparameter sensitivity and often necessitates extra tuning.

A complementary line of work reduces memory by computing the second moment at a coarser granularity. Adam-mini (Zhang et al., 2025b) (along with related methods such as BAGM (Zheng & Kwok, 2019), Adalayer (Zhao et al., 2025), NovoGrad (Ginsburg et al., 2020)) aggregates variance statistics over large blocks. However, applying block-wise statistics only to the second moment changes the update dynamics in ways that deviate from the sign-based behavior associated with Adam's performance characteristics (see Remark 2.1). AdamS (Zhang et al., 2025a) simplifies normalization by assuming the local geometry is governed by $(L_0, L_1)$-smoothness, replacing the second moment with a momentum-based term. Despite its design goals, we observed that this formulation lacks stability in practice, leading to divergence in our pre-training setups when using standard AdamW hyperparameters.

The emerging Muon optimizer (Jordan et al., 2024) achieves faster iteration-wise convergence without second moments, but its Newton-Schulz-based update introduces additional communication and computation overhead. While infras-

[1]The Chinese University of Hong Kong (Shenzhen) [2]Meituan LongCat Team. Correspondence to: Shi Pu <pushi@cuhk.edu.cn>, Jiaqi Zhang <zhangjiaqi39@meituan.com>.

*Proceedings of the 43rd International Conference on Machine Learning*, Seoul, South Korea. PMLR 306, 2026. Copyright 2026 by the author(s).

*Table 1.* **Structural and efficiency comparison of different optimizers.** "Compute Overhead" is classified as Low for element-wise or block-reduction updates (like BAS/Adam-mini), versus High for matrix-factorization-based updates (Muon). "States Stored" denotes whether first ($m$) and second ($v$) moments are maintained and at what granularity. "Opt. State Memory" reports the optimizer state footprint relative to AdamW. "Drop-in" indicates that the algorithm can transfer AdamW-style hyperparameters while maintaining stable and competitive convergence in our tested settings. Additional details are provided in Appendix B.

| OPTIMIZER | UPDATE TYPE | COMPUTE OVERHEAD | STATES STORED | OPT. STATE MEMORY | 1-BIT COMM. | DROP-IN? |
|---|---|---|---|---|---|---|
| ADAMW | SIGN-LIKE | LOW | $m, v$ (ELEM.) | $1\times$ | NO | YES |
| LION | SIGN | LOW | $m$ (ELEM.) | $1/2\times$ | YES | NO |
| ADAM-MINI | SGD-LIKE[†] | LOW | $m$ (ELEM.), $v$ (BLOCK) | $1/2\times$ | NO | PARTIAL[‡] |
| ADAMS | SIGN-LIKE | LOW | $m$ (ELEM.) | $\approx 1/2\times$ | NO | PARTIAL[‡] |
| MUON | ORTHOGONAL | HIGH | $m$ (ELEM.) | $1/2\times$ | NO | NO |
| **BAS (OURS)** | **SIGN-LIKE** | **LOW** | $m$ **(ELEM.)**, $v$ **(BLOCK)** | $\approx 1/8\times$ | **YES** | **YES** |

[†] Adam-mini exhibits dynamics closer to normalized SGDM in coarse-block limits; see Remark 2.1. [‡] Although the papers report that their methods can inherit AdamW hyperparameters, we observed instability or divergence in our pre-training transferability tests.

tructure optimizations can mask these costs, they significantly complicate deployment. Moreover, Muon exhibits dynamics that differ substantially from Adam; existing evidence suggests a potential optimizer mismatch when fine-tuning or continuing training from Adam-pretrained models (Liu et al., 2025).

In this paper, we introduce **Block Adaptive Signum (BAS)**, a novel optimizer that bridges the gap between the adaptivity of Adam and the memory efficiency of sign-based methods. BAS partitions parameters into blocks and computes a single adaptive scale per block. This design preserves Adam-like variance adaptation at a block-wise granularity while reducing second-moment storage to a single scalar per block. Furthermore, the sign-based nature of the BAS update is robust to quantization, enabling us to store the first moment ($m$) in FP8 format without performance degradation. Crucially, BAS structurally aligns with Adam (see Figure 4 for empirical validation of this alignment), and our sensitivity studies in Appendix E.1 show that AdamW-style hyperparameters transfer well across the tested regimes. The structural and efficiency advantages of BAS relative to existing state-of-the-art optimizers are summarized in Table 1.

Our contributions are as follows:

- **Method & Efficiency:** We propose Block Adaptive Signum (BAS), which unifies the variance-adaptivity of Adam with the memory-efficiency of SignSGD. By leveraging sign-based updates, BAS stores the first moment in FP8 and largely discards the second moment without performance loss, reducing the total optimizer state footprint to just **12.5% of AdamW**.

- **Theoretical Guarantee:** We establish the theoretical convergence of BAS under bounded-gradient and bounded-variance assumptions, proving that it converges to the neighborhood of a stationary point at the rate of $\mathcal{O}(1/\sqrt{T})$.

- **1-Bit Communication**: We introduce BAS-MV, a distributed variant that leverages sign-based majority voting to reduce communication volume compared to mixed-precision Distributed Data Parallel (DDP). Validated on the 0.5B regime, including added data-parallel scaling and wall-clock studies in Appendix B.4, it accelerates training in bandwidth-constrained environments.

- **Scalability & Performance:** We validate BAS through extensive empirical evaluation, including pre-training a 1.5B model on 100B tokens and industrial-scale Supervised Fine-Tuning (SFT) of models up to 32B parameters. Our results show that BAS is a promising drop-in candidate of AdamW, maintaining Adam-level performance across diverse large-scale training settings.

**Paper Organization.** Due to space constraints, we defer a detailed discussion of related works to Appendix A. The remainder of the paper presents the proposed method (Section 2), theoretical analysis (Section 3), and experimental results (Section 4).

## 2. The Proposed Method

We propose **Block Adaptive Signum (BAS)**, as detailed in Algorithm 1, which bridges the gap between the element-wise adaptivity of Adam and the memory efficiency of Signum. The core design principle is to adapt to variance in an Adam-like fashion while utilizing a negligible amount of second-moment storage. By partitioning the model into blocks, the algorithm tunes the step size based on the average signal-to-noise level at a block-wise granularity (see Sec. 2.3 for the detailed block partitioning scheme).

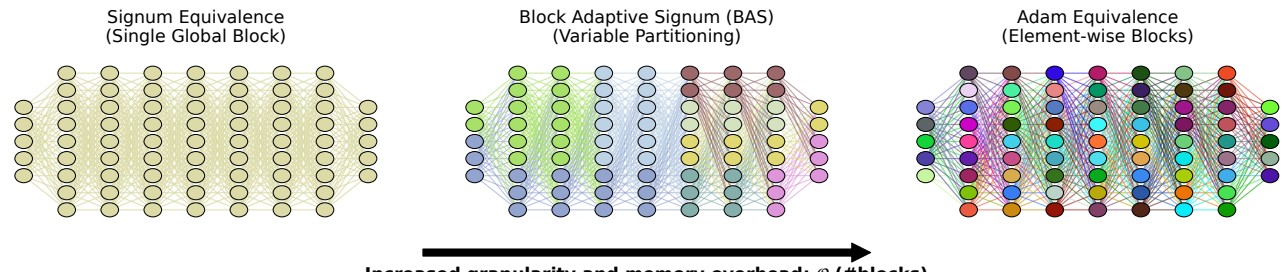

**Increased granularity and memory overhead:** $\mathcal{O}$ **(#blocks)**

*Figure 1.* **Schematic of the Block Adaptive Signum (BAS) spectrum.** The algorithm partitions parameters $x$ into disjoint blocks. **Left** (Signum Equivalence): All parameters form a single block, sharing one scale factor (low memory, low adaptivity). **Middle** (BAS): Flexible block sizes allow a trade-off between granularity and memory overhead. **Right** (Adam Equivalence): Element-wise blocking, where every parameter has a unique adaptive scale (high memory, max adaptivity). **Colors represent unique adaptive scales** $\alpha_t^b$.

## 2.1. Problem Setup

We consider the general unconstrained optimization problem

$$\min_{x \in \mathbb{R}^d} f(x). \tag{1}$$

In LLM training, the objective $f(x) = \mathbb{E}_{\xi \sim \mathcal{D}} F(x, \xi)$, where $F$ is the loss function and $\mathcal{D}$ is the data distribution. A batch gradient $g(x)$ is used to estimate $\nabla f(x)$.

**Notation.** We use $[n]$ to denote the set $\{1, \ldots, n\}$. We treat vectors as column vectors; $\mathbf{1}_d$ denotes the all-ones vector in $\mathbb{R}^d$. For operators, $\langle \cdot, \cdot \rangle$ is the inner product, $\odot$ is the element-wise (Hadamard) product, and $\mathrm{Sign}(\cdot)$ is the element-wise sign function. We denote the Frobenius norm by $\| \cdot \|_F$, the Euclidean norm by $\| \cdot \|_2$, and the $\ell_1$ norm by $\| \cdot \|_1$. $\mathrm{vec}(A)$ denotes the vectorization of $A$. Throughout, "parameter" refers to a Torch tensor (e.g., a weight matrix) treated as a block, while "element" refers to a single scalar entry. The model parameters $x \in \mathbb{R}^d$ are partitioned into $N$ disjoint blocks, written as $x = \{x_\mathbf{1}, x_\mathbf{2}, ..., x_\mathbf{N}\}$, where $x_\mathbf{i} \in \mathbb{R}^{d_\mathbf{i}}$. We use a bold subscript $\cdot_\mathbf{i}$ for block-level quantities (e.g., $g_\mathbf{i}^t$) and a standard subscript $\cdot_j$ for scalar elements. Finally, $g^t$, $m^t$, and $v^t$ denote the stochastic gradient, first moment, and second moment at step $t$, respectively; in distributed contexts, $P$ denotes the number of agents.

## 2.2. BAS: A Continuum Between Adam and Signum

The core flexibility of BAS lies in its block partitioning strategy, which governs the granularity of variance adaptation. By varying the size of these blocks, BAS defines a continuum of behavior between two classical optimization regimes (Figure 1):

- **Adam Equivalence**: When each parameter element forms its own block, BAS reduces to Adam.

- **Signum Equivalence**: When the entire parameter set is treated as a single block, BAS becomes Signum with an adaptive step size.

At each iteration, BAS estimates a block-wise step size scale $\alpha_\mathbf{i}^t$ and applies a *uniform-length update to every element within the block*. That is, all coordinates in block $\mathbf{i}$ move the same distance, differing only in direction through the element-wise sign of the block's first moment. This structure is what allows BAS to interpolate smoothly between Adam-style element-wise adaptivity and Signum-style global normalization.

The scale $\alpha_\mathbf{i}^t$ follows an Adam-style construction: it depends on the ratio between the block-wise first-moment norm $\|m_\mathbf{i}^t\|_F$ and the square root of the Exponential Moving Average (EMA) of the squared gradient norm within the block $\|g_\mathbf{i}^t\|_F^2$. We use the Frobenius norm because it is inexpensive to compute and corresponds to the $l_2$ norm of the matrix's singular-value vector. As in Adam, this mechanism decreases the effective learning rate when the block's gradient variance is high and vice versa.

Because the scale is defined using similar ingredients as AdamW, BAS can transfer its hyperparameters, including the learning rate $\eta$, $\beta$ values, and $\epsilon$, in the tested pre-training and fine-tuning regimes. This makes BAS a practical drop-in candidate for existing AdamW training pipelines, while broader no-retuning guarantees remain an empirical question.

Finally, BAS stores only one second-moment value per block, so the optimizer-state memory scales with the number of blocks $N \ll d$, effectively removing the second-moment overhead. In our experiments, this design attains Adam-level performance under transferred AdamW-style recipes while using substantially less memory.

*Remark* 2.1 (Comparison with Adam-mini). BAS is conceptually related to Adam-mini (Zhang et al., 2025b), which also utilizes block-wise second moments. However, they differ fundamentally in update dynamics. Interpreted through a sign-update perspective, Adam-mini is

$$x_j^{t+1} - x_j^t = -\eta \frac{m_j^t}{\sqrt{\bar{v}_\mathbf{i}^t}} = -\eta \frac{|m_j^t|}{\sqrt{\bar{v}_\mathbf{i}^t}} \mathrm{Sign}(m_j^t),$$

---

**Algorithm 1** Block Adaptive Signum (BAS)

---

**Require:** $\eta, \beta_1, \beta_2, \epsilon, \lambda$, and partition $x = \{x_1, \ldots, x_N\}$
1: Initialize $m^0 \leftarrow 0, v^0 \leftarrow 0$
2: **while** not converged **do**
3:     $t \leftarrow t + 1$, compute batch gradients $g^t$
4:     **for** each block $\mathbf{i} \in \{\mathbf{1}, \ldots, \mathbf{N}\}$ **do**
5:        $m_{\mathbf{i}}^t \leftarrow \beta_1 m_{\mathbf{i}}^{t-1} + (1 - \beta_1) g_{\mathbf{i}}^t$     *// $vec(m_{\mathbf{i}}^t) \in \mathbb{R}^{d_{\mathbf{i}}}$*
6:        $v_{\mathbf{i}}^t \leftarrow \beta_2 v_{\mathbf{i}}^{t-1} + (1 - \beta_2) \|g_{\mathbf{i}}^t\|_F^2$     *// $v_{\mathbf{i}}^t \in \mathbb{R}$*
7:        $\hat{m}_{\mathbf{i}}^t \leftarrow \frac{\|m_{\mathbf{i}}^t\|_F}{1 - \beta_1^t}, \quad \hat{v}_{\mathbf{i}}^t \leftarrow \frac{v_{\mathbf{i}}^t}{1 - \beta_2^t}$     *// Bias Correction*
8:        $\alpha_{\mathbf{i}}^t \leftarrow \hat{m}_{\mathbf{i}}^t / (\sqrt{\hat{v}_{\mathbf{i}}^t} + \epsilon)$     *// Scalar Adaptive Scale*
9:        $x_{\mathbf{i}}^{t+1} \leftarrow x_{\mathbf{i}}^t (1 - \eta\lambda) - \eta\alpha_{\mathbf{i}}^t \mathrm{Sign}(m_{\mathbf{i}}^t)$     *// Update*
10:    **end for**
11: **end while**

---

where $\bar{v}_{\mathbf{i}}$ denotes the average second moment of the block. Adam-mini therefore keeps coordinate-dependent update magnitudes $|m_j^t|/\sqrt{\bar{v}_{\mathbf{i}}^t}$ within each block. BAS instead applies a shared block magnitude $\alpha_{\mathbf{i}}^t$ and uses $\mathrm{Sign}(m_j^t)$ only for the coordinate-wise direction. In the one-block limit, Adam-mini degenerates to normalized SGDM, while BAS becomes Signum with an adaptive global scale. This uniform sign-based geometry is the main conceptual difference we emphasize. Prior work has shown that sign-based updates are an important ingredient behind why Adam is helpful for training language models (Kunstner et al., 2024), and our experiments suggest that preserving this geometry improves AdamW-style hyperparameter transferability relative to Adam-mini in the tested setting.

*Remark* 2.2 (Comparison with Muon). Although the recent Muon (Jordan et al., 2024) optimizer has shown promising results, its update rule is fundamentally different from ours, relying on Newton-Schulz iterations that introduce substantial computation and communication overhead in distributed settings. Existing evidence suggests that Muon can introduce an optimizer mismatch when fine-tuning or continuing the pre-training of Adam-based models (Liu et al., 2025). In contrast, BAS more closely follows Adam's sign-times-scale structure and is intended for Adam-style training recipes. Beyond these differences, the current Muon update, like Signum, lacks variance adaptation. While block-adaptive scaling could in principle be incorporated into Muon, such a combination does not yield the desired behavior in practice; we provide further discussion in Appendix B.5.

### 2.3. Implementation Details for LLM Training

**FP8 Moment Storage.** While traditional optimizers require FP32 moments to ensure update fidelity, BAS does not. Its sign-based direction and block-wise scaling are inherently robust to quantization, allowing FP8 moment storage without performance degradation. We therefore adopt FP8

moment storage in all experiments. Compared to AdamW, which stores element-wise FP32 $m, v$ buffers, BAS requires only 12.5% of the optimizer-state memory by almost discarding $v$ and storing $m$ in FP8, giving a $4\times$ reduction in optimizer-state footprint compared to other methods. Appendix D.2.3 further clarifies the precision boundary: FP8 matches BF16/FP32 in our tested setup, while naive 6-bit storage degrades and naive 4-bit storage is unstable.

**Block Partitioning Scheme.** BAS does not require the highly fine-grained partition schemes utilized by Adam-mini. For the embedding and output layers, we require row-wise partitioning—treating each vocabulary entry as a block—because their gradients are (numerically) sparse. For all other parameters, there are no strict constraints.

For our main experiments, we utilize an intuitive **parameter-wise partition**, where each Torch parameter is treated as a block. Block sizes span from $\mathcal{O}(10^3)$ LayerNorm vectors to $\mathcal{O}(10^6 - 10^8)$ projection matrices, and the majority of parameters lie in blocks exceeding $10^6$ elements. In distributed frameworks like FSDP, parameters are sharded across devices. BAS remains highly efficient in this setting because computing the block-wise scale $\alpha_{\mathbf{i}}^t$ requires reducing only two scalars per block (the square Frobenius norm of each shard's first moment $\|m_{\mathbf{i}}^t\|_F^2$ and batch gradient $\|g_{\mathbf{i}}^t\|_F^2$), incurring negligible communication overhead (see Appendix D.1.1) compared to the gradient exchange.

A more convenient scheme is the **shard-wise partition**, where each local shard is treated as a block. This incurs no communication overhead and offers substantial flexibility (see Appendix D.1.2). Because shard-wise partitioning yields a finer block structure that more closely resembles AdamW's element-wise adaptivity, it behaves stably in practice. However, shard boundaries depend on the specific sharding implementation and data-parallel world size, leading to small cross-framework variations. For reproducibility, we therefore adopt the parameter-wise partition as the default in our main experiments.

### 2.4. Communication-Efficient Variant: BAS-MV

The sign-based structure of BAS naturally enables a communication-efficient distributed variant, presented in Algorithm 2: **BAS with Majority Vote (BAS-MV)**. In this setting, each worker computes its own block-wise adaptive scales locally. Workers then communicate the signs of their first-moment estimates (via 1-bit majority vote) and their scalar block scales (via averaging). Because the scalar scale communication is negligible ($N \ll d$), BAS-MV achieves up to a $16\times$ reduction in communication volume compared to standard DDP with BF16 gradients. We define the relative communication budget as $B_{\mathrm{rel}}(t) = t \cdot C_{\mathrm{alg}}/C_{\mathrm{DDP}}$, where $C_{\mathrm{alg}}$ is the per-step communication volume of the optimizer under comparison. Under this metric, BAS-MV

can make more training progress under the same communication budget; Appendix B.4 adds fixed-token, data-parallel scaling, and wall-clock evidence.

While BAS-MV provides strong communication savings, it also introduces practical constraints. In particular, BAS-MV is *not* compatible with ZeRO-style optimizer-state sharding, since each worker must maintain its own local first and second moments to compute block-wise scales. Nevertheless, for moderately small data-parallel sizes, BAS-MV attains memory usage comparable to ZeRO-2; a detailed analysis is provided in Appendix B.4. Consequently, BAS-MV is most suitable for communication-constrained data-parallel LLM training where the model is moderate enough to fit with replicated optimizer states.

---

**Algorithm 2** BAS with Majority Vote (BAS-MV)

---

**Require:** $\eta, \beta_1, \beta_2, \epsilon, \lambda, x = \{x_\mathbf{1}, \ldots, x_\mathbf{N}\}$, and $P$ agents
1: Initialize local $m^{0,j} \leftarrow 0$, $v^{0,j} \leftarrow 0$ for agents $j \in [P]$
2: **while** not converged **do**
3:     $t \leftarrow t + 1$; Agents compute local gradients $g^{t,j}$
4:     **for** each block $\mathbf{i} \in \{\mathbf{1}, \ldots, \mathbf{N}\}$ **do**
5:        **Each agent** $j \in [P]$ **locally computes:**
6:          $m_\mathbf{i}^{t,j} \leftarrow \beta_1 m_\mathbf{i}^{t-1,j} + (1 - \beta_1) g_\mathbf{i}^{t,j}$
7:          $v_\mathbf{i}^{t,j} \leftarrow \beta_2 v_\mathbf{i}^{t-1,j} + (1 - \beta_2) \|g_\mathbf{i}^{t,j}\|_F^2$
8:          $\hat{m}_\mathbf{i}^{t,j} \leftarrow \frac{\|m_\mathbf{i}^{t,j}\|_F}{1 - \beta_1^t}$,   $\hat{v}_\mathbf{i}^{t,j} \leftarrow \frac{v_\mathbf{i}^{t,j}}{1 - \beta_2^t}$
9:          $\alpha_\mathbf{i}^{t,j} \leftarrow \hat{m}_\mathbf{i}^{t,j}/(\sqrt{\hat{v}_\mathbf{i}^{t,j}} + \epsilon)$      *// Local Scale*
10:     $\bar{\alpha}_\mathbf{i}^t \leftarrow \frac{1}{P} \sum_{j=1}^{P} \alpha_\mathbf{i}^{t,j}$       *// Aggregate Scales*
11:     $V_\mathbf{i}^t \leftarrow \sum_{j=1}^{P} \text{Sign}(m_\mathbf{i}^{t,j})$       *// Sign Voting*
12:     $x_\mathbf{i}^{t+1} \leftarrow x_\mathbf{i}^t(1 - \eta\lambda) - \eta\bar{\alpha}_\mathbf{i}^t\text{Sign}(V_\mathbf{i}^t)$    *// Update*
13:     **end for**
14: **end while**

---

## 3. Convergence Analysis

We analyze the convergence properties of BAS for general non-convex objectives under bounded-gradient and bounded-variance assumptions, focusing on the core algorithm without weight decay or bias correction. These assumptions can be restrictive relative to the strongest recent Adam analyses, but provide a first guarantee for BAS. The following assumptions form the basis of our results.

**Assumption 3.1** (L-Smoothness and lower-boundedness). The function $f$ is $L$-smooth. For all $x, y \in \mathbb{R}^d$: $f(y) \leq f(x) + \langle\nabla f(x), y - x\rangle + \frac{L}{2}\|y - x\|_2^2$, and $f$ is bounded from below by $f^*$.

**Assumption 3.2** (Bounded Variance). The stochastic gradient is unbiased, $\mathbb{E}[g^t] = \nabla f(x^t)$, and has bounded variance: $\mathbb{E}[\|g^t - \nabla f(x^t)\|_2^2] \leq \sigma^2$. Also, the noise vectors $\zeta^t := g^t - \nabla f(x^t)$ are independent across time steps.

**Assumption 3.3** (Bounded Gradient). The gradient of the function $f$ is bounded: $\|\nabla f(x)\|_1 \leq C_g$ for all $x$.

The bounded-gradient assumption has been used in prior analyses of Adam-type methods (Reddi et al., 2019; Défossez et al., 2022). For the ease of analysis, we rewrite BAS in a compact way. Since the Frobenius norm of a matrix coincides with the $l_2$ norm of its vectorization, we flatten each block into a vector.

The resulting BAS update can be written as

$$x^{t+1} = x^t - \eta(\alpha^t \odot \text{Sign}(m^t)),$$

where $\alpha^t = [\alpha_\mathbf{1}^t \cdot \mathbf{1}_{d_1}^\top, \alpha_\mathbf{2}^t \cdot \mathbf{1}_{d_2}^\top, .., \alpha_\mathbf{N}^t \cdot \mathbf{1}_{d_\mathbf{N}}^\top]^\top \in \mathbb{R}^d$. With this compact notation in place, we now proceed to derive the convergence theorem.

**Theorem 3.4.** *Under Assumptions 3.1-3.3, when $\beta_2 \geq \beta_1$, BAS converges as follows:*

$$\frac{\sum_{t=0}^{T-1} \mathbb{E}\|\nabla f(x^t)\|_1^2}{T} \leq C_{total}\left[\frac{\Delta_f}{\eta T} + \frac{Ld}{2}C_\alpha^2\eta\right.$$

$$\left. + (2C_\alpha + C_g/\epsilon)\left(\frac{C_g}{T\bar{\beta}_1} + \frac{L\eta C_\alpha d}{\bar{\beta}_1} + \sigma\sqrt{d\bar{\beta}_1}\right)\right],$$

*where $\Delta_f = f(x^0) - f^*$, $C_\alpha = \sqrt{\frac{1-\beta_1}{1-\beta_2}}$, $\bar{\beta}_1 = 1 - \beta_1$, and $C_{total} = dN\left(\sqrt{C_g^2 + \sigma^2} + \epsilon\right)$.*

We next state a corollary that provides the convergence rate under a specific stepsize.

**Corollary 3.5.** *Let the stepsize be set as $\eta = \frac{1}{\sqrt{T}}$. Under the assumptions of Theorem 3.4, BAS converges to a neighborhood of a stationary point with rate $O(1/\sqrt{T})$:*

$$\frac{1}{T}\mathbb{E}\sum_{t=0}^{T-1}\|\nabla f(x^t)\|_1^2 \leq \frac{\mathcal{C}_{\text{rate}}}{\sqrt{T}} + \mathcal{C}_{\text{nbd}},$$

*where $\mathcal{C}_{\text{nbd}} \sim \mathcal{O}(\sqrt{1 - \beta_1})$, and the formal definitions of $\mathcal{C}_{\text{rate}}$ and $\mathcal{C}_{\text{nbd}}$ are deferred to Appendix C.*

*Remark* 3.6. Convergence to a neighborhood of a stationary point is a standard outcome in the analysis of adaptive optimizers with constant hyperparameters (Wang et al., 2024). The radius of this neighborhood, $\mathcal{C}_{\text{nbd}}$, is determined by the system's noise floor ($\sigma$) and the momentum coefficient $(1 - \beta_1)$. Thus the method approaches the optimum quickly, but its final accuracy is limited by gradient noise and the fixed stepsize, mirroring the behavior of SGD and Adam.

## 4. Experiments

We evaluate the proposed BAS method across two distinct training regimes: pre-training and SFT.

### 4.1. Experimental Setup

**Baselines.** We compare BAS with four primary baselines: 1. AdamW (Loshchilov & Hutter, 2019), the standard choice

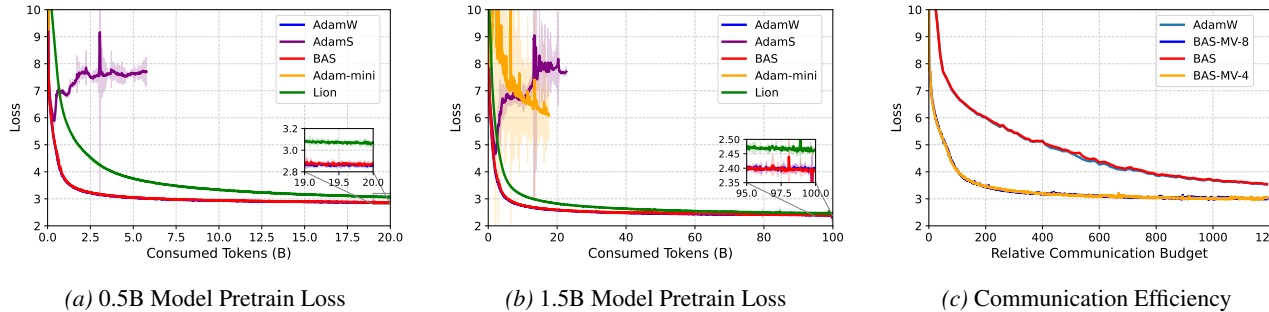

*(a) 0.5B Model Pretrain Loss*      *(b) 1.5B Model Pretrain Loss*     *(c) Communication Efficiency*

*Figure 2.* **Pre-training Stability and Efficiency Analysis. (a, b)** Training loss on Qwen2.5 (0.5B & 1.5B) over 20B & 100B tokens. BAS (red) matches the convergence trajectory of AdamW (blue) under transferred AdamW-style hyperparameters while Adam-mini and AdamS diverge, and Lion converges slower. **(c)** Convergence speed versus relative communication budget for the 0.5B model. BAS-MV (orange and blue) achieves lower training loss for the same communication budget compared to baselines (numbers indicate the number of agents); Appendix B.4 gives the precise budget definition and further wall-clock results.

for large-scale LLM training; 2. Lion (Chen et al., 2023), a widely adopted sign-based optimizer known for memory efficiency; 3. Adam-mini (Zhang et al., 2025b) and AdamS (Zhang et al., 2025a), two recent Adam variants designed to reduce memory overhead. We exclude Muon (Jordan et al., 2024) for the reasons discussed in Remark 2.2. Unless explicitly noted as **BAS-MV**, all references to **BAS** denote Algorithm 1 with param.-wise partition.

**Implementation & Infrastructure.** All experiments are implemented in PyTorch using the DeepSpeed framework (ZeRO-2 for models up to 7B, ZeRO-3 for the 32B model). Each training run is executed on a single node with 8 NVIDIA GPUs. All main experiments are repeated over three random seeds. FP8 storage uses the e5m2 format with a fixed scaling factor of 512 for all main experiments.

**Datasets & Models.** Our pre-training experiments use Qwen2.5 (Qwen et al., 2025) models at 0.5B (20B tokens) and 1.5B (100B tokens), trained on FineWeb-Edu (Penedo et al., 2024). For SFT, we fine-tune Olmo3 (Olmo et al., 2025) models of sizes 7B and 32B on the Dolci-Instruct-SFT dataset to produce instruction-tuned variants.

**Hyperparameters**. Across all experiments, we fix the momentum coefficients to $\beta = (0.9, 0.95)$ for every optimizer, a setting commonly used in recent LLM training pipelines (Brown et al., 2020; Penedo et al., 2024). Our main pre-training comparison is primarily a hyperparameter-transferability test: we tune AdamW, transfer its recipe to BAS, Adam-mini, and AdamS, and separately tune Lion because its standard recipe differs substantially. We also report additional learning-rate, beta, and weight-decay sweeps in Appendix E.1 to distinguish transferability from tuned optimizer quality. For Olmo3 SFT, we do not search for AdamW hyperparameters and instead directly adopt the AdamW hyperparameters reported in the Olmo3 report without additional tuning. We apply a warm-up phase over 3% of tokens,

followed by a constant learning rate during pre-training and a linear decay schedule during SFT. The sequence lengths for pre-training and SFT are 4K and 32K, respectively. For pre-training, the 0.5B model uses a global batch size of 256, and the 1.5B model uses a global batch size of 1024. For SFT, the global batch sizes are 32 for the 7B model and 128 for the 32B model. Additional details are provided in Appendix D.2.

### 4.2. Pre-training Results on Qwen2.5

We begin by evaluating the optimization stability of BAS during the pre-training phase. Figure 2a and 2b present the training loss trajectories for the 0.5B and 1.5B parameter variants. Notably, the 1.5B model was trained on 100B tokens; this represents a significantly more extensive training regime than the 20B tokens typically reported for models of this scale in prior literature.

**Robustness Comparison.** A notable finding is the difference in AdamW-hyperparameter transferability in this setting. Both Adam-mini and AdamS diverged early in the Qwen2.5 runs when given the transferred AdamW recipe. In contrast, BAS maintained stability throughout training, closely tracking the convergence trajectory of AdamW. We do not interpret this as a claim that Adam-mini or AdamS are generally unstable under their own tuned recipes; Appendix E.1 reports additional retuning results.

**Convergence Speed.** While Lion remained stable, it exhibited slower convergence compared to AdamW and BAS in terms of token efficiency. BAS achieves a final loss effectively identical to AdamW in these runs, suggesting that block-wise adaptive scaling captures enough variance information for efficient training under the tested recipes.

**On Communication Efficiency.** Figure 2c compares BAS-MV with the baselines under a fixed relative communication

*Table 2.* **Main Results on the Olmo 3 Evaluation Suite.** BAS is evaluated against four baselines in the 7B regime and against AdamW in the 32B regime across 19 benchmarks (10 displayed here; full results in Table 10). BAS (highlighted in gray) matches AdamW's performance while surpassing other memory-efficient optimizers.

| | **7B REGIME** | | | | | **32B REGIME** | |
| BENCHMARK | ADAMW | LION | ADAMS | ADAM-MINI | **BAS (OURS)** | ADAMW | **BAS (OURS)** |
| --- | --- | --- | --- | --- | --- | --- | --- |
| ***Math*** | | | | | | | |
| MATH | 58.6 | 56.5 | 57.8 | 57.2 | **59.2** | **69.2** | 69.1 |
| AIME 2024 | 5.2 | 4.1 | **5.4** | 4.2 | 4.9 | 9.0 | **9.2** |
| ***Reasoning*** | | | | | | | |
| BIGBENCHHARD | 49.5 | 47.5 | 48.6 | 48.1 | **50.3** | 66.7 | **67.9** |
| ZEBRALOGIC | **14.5** | 12.9 | 13.5 | 13.7 | 13.7 | 25.4 | **26.7** |
| ***Coding*** | | | | | | | |
| HUMANEVALPLUS | **65.2** | 60.8 | 62.9 | 63.2 | 63.7 | 74.3 | **75.1** |
| MBPP+ | **51.5** | 46.5 | 49.6 | 49.2 | 50.2 | **56.9** | 56.7 |
| ***Instruction Following*** | | | | | | | |
| IFEVAL | **76.0** | 72.5 | 73.8 | 73.5 | 75.2 | **85.2** | 84.1 |
| IFBENCH | 25.3 | 24.1 | 24.8 | 25.2 | **26.5** | 27.7 | **28.2** |
| ***Knowledge & QA*** | | | | | | | |
| MMLU | 66.0 | 63.8 | 64.5 | 64.2 | **66.2** | **78.5** | 78.1 |
| POPQA | **15.7** | 14.5 | 14.9 | 14.7 | 15.2 | **23.1** | 21.9 |
| ***Chat & Tool Use*** | | | | | | | |
| ALPACAEVAL 2 LC | 20.2 | 18.5 | 19.1 | 18.8 | **21.3** | 40.7 | **41.5** |
| SIMPLEQA | 71.0 | 70.5 | 71.2 | 70.8 | **73.4** | **84.2** | 83.5 |

budget, assuming DDP with BF16 gradients and thus a $16\times$ communication-volume advantage for BAS-MV before negligible scalar reductions. With 4 and 8 agents, BAS-MV converges faster under the same communication budget. Appendix B.4 adds fixed-token DP scaling up to 128 agents and a wall-clock study on PCIe-connected RTX 3090 GPUs, while also making clear that BAS-MV has gradual degradation as DP increases.

### 4.3. Large-Scale Instruction Fine-Tuning

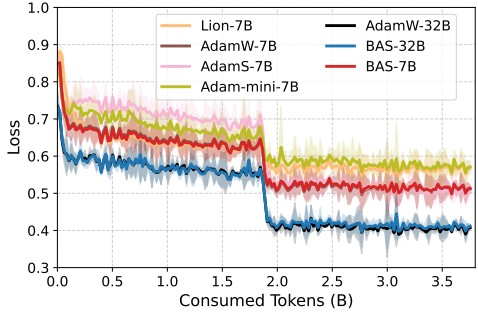

*Figure 3.* Training Loss for SFT on Olmo-7B and Olmo-32B.

To validate BAS in a production setting, we apply it to the supervised instruction tuning of Olmo3-7B and Olmo3-32B, transforming the base checkpoints into instruction-tuned

models (Olmo et al., 2025). To ensure the results reflect real-world practice, we train on more than 3B tokens, consistent with production-scale instruction-tuning regimes rather than smaller academic proxies.

**Benchmarking 7B Performance.** Table 2 reports downstream performance across standard benchmarks, spanning Math, Coding, and Reasoning. Two key trends emerge. First, BAS closely matches the performance of the full-memory AdamW optimizer (Loshchilov & Hutter, 2019) across categories in this SFT setting, indicating that our block-wise second moment approximation preserves downstream quality. Second, BAS compares favorably with the tested memory-efficient baselines, including Lion (Chen et al., 2023), AdamS (Zhang et al., 2025a), and Adam-mini (Zhang et al., 2025b) in the 7B regime. These comparisons should be read together with the sensitivity studies in Appendix E.1: our central claim is strong AdamW-style transferability and memory savings, not universal dominance under every possible tuning budget.

**Scalability to 32B Parameters.** To assess the stability of BAS at scale, we fine-tuned the Olmo3-32B model. Due to the high computational cost, we limited the baseline comparison to AdamW. As illustrated in Figure 3, BAS successfully scales to the 32B parameter regime, exhibiting a training loss trajectory that closely tracks AdamW. This result indicates that BAS's coarse-grained (block-wise) signal-to-noise

estimation remains robust at large scale, consistent with the stability observed in our pretraining experiments.

### 4.4. Ablation and Analysis

**Sensitivity to Design Choices.** We evaluate the sensitivity of BAS to key design choices, including block partitioning granularity (Layer-wise, Parameter-wise, Shard-wise), momentum precision (FP32, BF16, FP8), and norm metric (Frobenius, Spectral, Nuclear). Figure 5 shows that BAS is generally robust to block size, though excessively coarse global sharding introduces mild degradation. We also find that the spectral norm yields slightly worse convergence. Given their stability and efficiency, we adopt the Frobenius norm and FP8 momentum storage in all main experiments.

**Structural Similarity of Trained Models.** Figure 6 compares the singular value spectra of models trained with different optimizers. BAS closely follows the spectral structure of AdamW, maintaining similar rank patterns and decay, whereas Lion, AdamS, and Adam-mini deviate more substantially. For Adam-mini, these deviations align with the update-geometry difference discussed in Remark 2.1, rather than with a claim that Adam-mini is generally unstable under all tuned settings.

**Why does BAS mimic AdamW?** BAS closely approximates AdamW across multiple metrics. Although one might expect AdamW's element-wise scales to be uniform within each parameter tensor, Figure 4 (Left & Middle) shows moderate intra-parameter heterogeneity. Nevertheless, BAS tracks the block-average scale with high accuracy, as reflected by the Block Coherence ratio remaining near 1.0 in Figure 4 (Right). This alignment helps explain why BAS matches AdamW in training loss, weight structure, and downstream performance.

## 5. Conclusion and Future Work

In this work, we introduced **Block Adaptive Signum (BAS)**, a memory-efficient optimizer that bridges the gap between the adaptivity of Adam and the efficiency of sign-based methods. By leveraging block-wise variance estimates and sign-based updates, BAS matches the convergence trajectory and downstream performance of AdamW in the tested large-scale language-model settings, scaling up to 32B parameters, while reducing optimizer state memory to approximately 12.5% of that required by AdamW. We further provided a first theoretical guarantee for BAS under bounded-gradient and bounded-variance assumptions, proving convergence to a neighborhood of a stationary point at a rate of $\mathcal{O}(1/\sqrt{T})$.

**Limitations and Future Work.** While BAS offers significant efficiency gains, it is not without limitations. First, our communication-efficient variant, BAS-MV, requires agents to maintain local moments, making it currently incompatible with ZeRO-style optimizer state sharding.Future work will explore system-level optimizations to reconcile these distributed paradigms. Second, our theoretical analysis relies on bounded-gradient and bounded-variance assumptions; relaxing these assumptions for nonsmooth or heavy-tailed settings is an important direction for future work. Third, our quantization evidence supports FP8 as a practical operating point, while naive 6-bit storage degrades and naive 4-bit storage is unstable in our tested setup. More aggressive quantization likely requires specialized low-bit scaling or mixed-precision designs.

## Acknowledgements

This work was supported in part by the National Key Research and Development Program of China under Grant 2025YFA1018800, in part by the National Natural Science Foundation of China (NSFC) under Grant 62373316 and Grant 62336005, and in part by the 1 + 1+1 CUHK-CUHK(SZ)-GDSTC Joint Collaboration Fund under Grant 2025A0505000049.

## Impact Statement

This paper presents work whose goal is to advance the field of Machine Learning by improving the memory efficiency of training Large Language Models (LLMs). By reducing the hardware requirements for pre-training and fine-tuning, our proposed method, BAS, contributes to the democratization of AI research. It enables practitioners with limited computational resources—such as academic laboratories and smaller institutions—to train or fine-tune state-of-the-art models that were previously accessible only to industrial labs with massive infrastructure.

Furthermore, by optimizing memory utilization and enabling training on fewer devices, this work aligns with "Green AI" initiatives aimed at reducing the energy consumption and carbon footprint of large-scale model training.

However, we acknowledge that lowering the barrier to training powerful models carries potential risks. Increased accessibility may facilitate the fine-tuning of models for malicious purposes (e.g., disinformation or malware generation) by actors with limited resources. We believe that the benefits of transparent, accessible, and efficient AI research outweigh these risks, but continued community effort toward model safety, alignment, and responsible release remains essential.

## References

Anil, R., Gupta, V., Koren, T., and Singer, Y. Memory-efficient adaptive optimization, 2019. URL https://arxiv.org/abs/1901.11150.

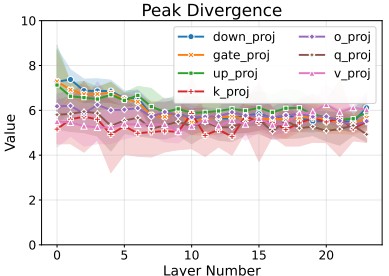 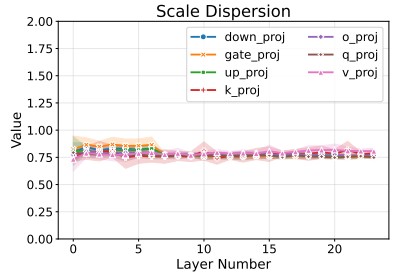 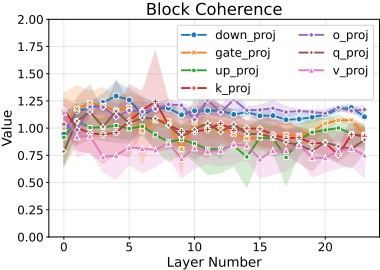

*Figure 4.* **Analysis of Intra-Block Update Dynamics in AdamW.** We visualize the parameter-wise training dynamics of AdamW and BAS at step 1000 (1B tokens). The standard deviation is calculated over a 10-step window, and we defer the detailed description on each metric to Appendix E.3 . **(Left) Peak Divergence:** The ratio of the maximum parameter update to the mean update within a parameter tensor. **(Middle) Scale Dispersion:** The coefficient of variation (CV) of the update scales. **(Right) Block Coherence:** The ratio of the block-wise step scale (BAS) to the local mean AdamW step scale. Values near 1.0 indicate high coherence between the two methods.

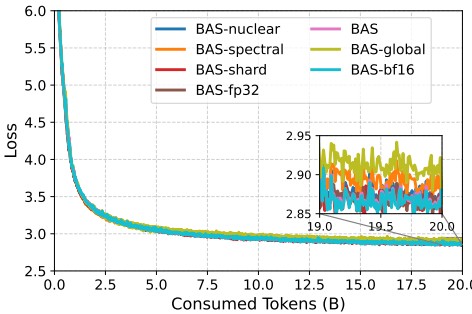

*Figure 5.* **Ablation Study.** Pretraining loss of Qwen2.5-0.5B under different design choices. The default BAS configuration uses parameter-wise partitioning with FP8 moment storage.

Bernstein, J., Wang, Y.-X., Azizzadenesheli, K., and Anandkumar, A. signsgd: Compressed optimisation for non-convex problems. In *International conference on machine learning*, pp. 560–569. PMLR, 2018.

Brown, T., Mann, B., Ryder, N., Subbiah, M., Kaplan, J. D., Dhariwal, P., Neelakantan, A., Shyam, P., Sastry, G., Askell, A., et al. Language models are few-shot learners. *Advances in neural information processing systems*, 33: 1877–1901, 2020.

Chen, X., Liang, C., Huang, D., Real, E., Wang, K., Pham, H., Dong, X., Luong, T., Hsieh, C.-J., Lu, Y., et al. Symbolic discovery of optimization algorithms. *Advances in neural information processing systems*, 36:49205–49233, 2023.

Dettmers, T., Lewis, M., Shleifer, S., and Zettlemoyer, L. 8-bit optimizers via block-wise quantization, 2022. URL https://arxiv.org/abs/2110.02861.

Défossez, A., Bottou, L., Bach, F., and Usunier, N. A simple convergence proof of adam and adagrad, 2022. URL https://arxiv.org/abs/2003.02395.

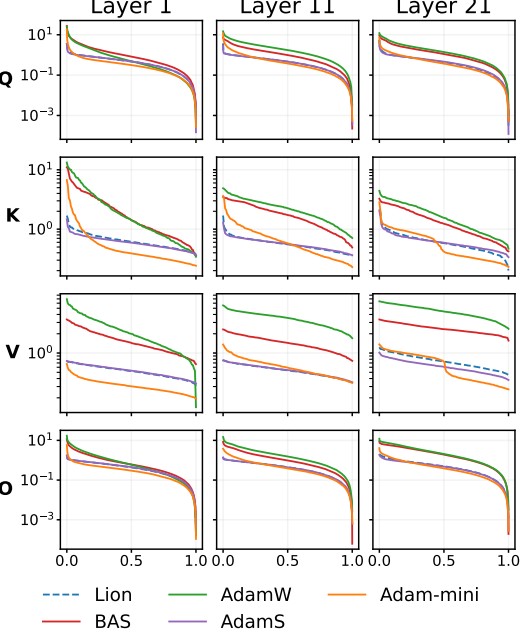

*Figure 6.* **Singular Value Spectra of Weight Matrices.** Singular value distributions of the Q, K, V, and O projections at Layers 1, 11, and 21 of **Qwen2.5-0.5B** trained with different optimizers. For AdamS and Adam-mini, we report results from their optimal learning rates, as they diverge under AdamW hyperparameters (Appendix E.1). **BAS** matches the spectral structure of **AdamW** more closely than other baselines.

Ginsburg, B., Castonguay, P., Hrinchuk, O., Kuchaiev, O., Lavrukhin, V., Leary, R., Li, J., Nguyen, H., Zhang, Y., and Cohen, J. M. Stochastic gradient methods with layer-wise adaptive moments for training of deep networks, 2020. URL https://arxiv.org/abs/1905.11286.

Jordan, K., Jin, Y., Boza, V., You, J., Cesista, F., Newhouse, L., and Bernstein, J. Muon: An optimizer for hidden layers in neural networks, 2024. URL https://kellerjordan.github.io/posts/muon/.

Kingma, D. P. and Ba, J. Adam: A method for stochastic optimization, 2017. URL https://arxiv.org/abs/1412.6980.

Kunstner, F., Milligan, A., Yadav, R., Schmidt, M., and Bietti, A. Heavy-tailed class imbalance and why adam outperforms gradient descent on language models. *Advances in Neural Information Processing Systems*, 37: 30106–30148, 2024.

Liu, H., Li, Z., Hall, D., Liang, P., and Ma, T. Sophia: A scalable stochastic second-order optimizer for language model pre-training. In *International Conference on Learning Representations*, volume 2024, pp. 1621–1650, 2024.

Liu, J., Su, J., Yao, X., Jiang, Z., Lai, G., Du, Y., Qin, Y., Xu, W., Lu, E., Yan, J., Chen, Y., Zheng, H., Liu, Y., Liu, S., Yin, B., He, W., Zhu, H., Wang, Y., Wang, J., Dong, M., Zhang, Z., Kang, Y., Zhang, H., Xu, X., Zhang, Y., Wu, Y., Zhou, X., and Yang, Z. Muon is scalable for llm training, 2025. URL https://arxiv.org/abs/2502.16982.

Loshchilov, I. and Hutter, F. Decoupled weight decay regularization, 2019. URL https://arxiv.org/abs/1711.05101.

Luo, Y., Ren, X., Zheng, Z., Jiang, Z., Jiang, X., and You, Y. CAME: Confidence-guided adaptive memory efficient optimization. In Rogers, A., Boyd-Graber, J., and Okazaki, N. (eds.), *Proceedings of the 61st Annual Meeting of the Association for Computational Linguistics (Volume 1: Long Papers)*, pp. 4442–4453, Toronto, Canada, July 2023. Association for Computational Linguistics. doi: 10.18653/v1/2023.acl-long.243. URL https://aclanthology.org/2023.acl-long.243/.

Olmo, T., :, Ettinger, A., Bertsch, A., Kuehl, B., Graham, D., Heineman, D., Groeneveld, D., Brahman, F., Timbers, F., Ivison, H., Morrison, J., Poznanski, J., Lo, K., Soldaini, L., Jordan, M., Chen, M., Noukhovitch, M., Lambert, N., Walsh, P., Dasigi, P., Berry, R., Malik, S., Shah, S., Geng, S., Arora, S., Gupta, S., Anderson, T., Xiao, T., Murray, T., Romero, T., Graf, V., Asai, A., Bhagia, A., Wettig, A., Liu, A., Rangapur, A., Anastasiades, C., Huang, C.,

Schwenk, D., Trivedi, H., Magnusson, I., Lochner, J., Liu, J., Miranda, L. J. V., Sap, M., Morgan, M., Schmitz, M., Guerquin, M., Wilson, M., Huff, R., Bras, R. L., Xin, R., Shao, R., Skjonsberg, S., Shen, S. Z., Li, S. S., Wilde, T., Pyatkin, V., Merrill, W., Chang, Y., Gu, Y., Zeng, Z., Sabharwal, A., Zettlemoyer, L., Koh, P. W., Farhadi, A., Smith, N. A., and Hajishirzi, H. Olmo 3, 2025. URL https://arxiv.org/abs/2512.13961.

Penedo, G., Kydlíček, H., Lozhkov, A., Mitchell, M., Raffel, C. A., Von Werra, L., Wolf, T., et al. The fineweb datasets: Decanting the web for the finest text data at scale. *Advances in Neural Information Processing Systems*, 37: 30811–30849, 2024.

Qwen, :, Yang, A., Yang, B., Zhang, B., Hui, B., Zheng, B., Yu, B., Li, C., Liu, D., Huang, F., Wei, H., Lin, H., Yang, J., Tu, J., Zhang, J., Yang, J., Yang, J., Zhou, J., Lin, J., Dang, K., Lu, K., Bao, K., Yang, K., Yu, L., Li, M., Xue, M., Zhang, P., Zhu, Q., Men, R., Lin, R., Li, T., Tang, T., Xia, T., Ren, X., Ren, X., Fan, Y., Su, Y., Zhang, Y., Wan, Y., Liu, Y., Cui, Z., Zhang, Z., and Qiu, Z. Qwen2.5 technical report, 2025. URL https://arxiv.org/abs/2412.15115.

Reddi, S. J., Kale, S., and Kumar, S. On the convergence of adam and beyond, 2019. URL https://arxiv.org/abs/1904.09237.

Schlotthauer, J., Kroos, C., Hinze, C., Hangya, V., Hahn, L., and Küch, F. Pre-training llms on a budget: A comparison of three optimizers, 2025. URL https://arxiv.org/abs/2507.08472.

Shazeer, N. and Stern, M. Adafactor: Adaptive learning rates with sublinear memory cost. In *International Conference on Machine Learning*, pp. 4596–4604. PMLR, 2018.

Vyas, N., Morwani, D., Zhao, R., Shapira, I., Brandfonbrener, D., Janson, L., and Kakade, S. Soap: Improving and stabilizing shampoo using adam for language modeling. In *International Conference on Learning Representations*, volume 2025, pp. 93423–93444, 2025.

Wang, B., Zhang, Y., Zhang, H., Meng, Q., Sun, R., Ma, Z.-M., Liu, T.-Y., Luo, Z.-Q., and Chen, W. Provable adaptivity of adam under non-uniform smoothness. In *Proceedings of the 30th ACM SIGKDD Conference on Knowledge Discovery and Data Mining*, pp. 2960–2969, 2024.

You, Y., Gitman, I., and Ginsburg, B. Large batch training of convolutional networks, 2017. URL https://arxiv.org/abs/1708.03888.

You, Y., Li, J., Reddi, S., Hseu, J., Kumar, S., Bhojanapalli, S., Song, X., Demmel, J., Keutzer, K., and Hsieh, C.-J. Large batch optimization for deep learning: Training bert in 76 minutes. *arXiv preprint arXiv:1904.00962*, 2019.

Zhang, H., Wang, B., and Chen, L. Adams: Momentum itself can be a normalizer for llm pretraining and post-training. In *Proceedings of the 2025 Conference on Empirical Methods in Natural Language Processing*, pp. 10730–10749, 2025a.

Zhang, Y., Chen, C., Li, Z., Ding, T., Wu, C., Kingma, D. D., Ye, Y., Luo, Z.-Q., and Sun, R. Adam-mini: Use fewer learning rates to gain more. In *International Conference on Learning Representations*, volume 2025, pp. 28033–28063, 2025b.

Zhao, J., Zhang, Z., Chen, B., Wang, Z., Anandkumar, A., and Tian, Y. Galore: Memory-efficient llm training by gradient low-rank projection, 2024. URL https://arxiv.org/abs/2403.03507.

Zhao, R., Morwani, D., Brandfonbrener, D., Vyas, N., and Kakade, S. Deconstructing what makes a good optimizer for language models, 2025. URL https://arxiv.org/abs/2407.07972.

Zheng, S. and Kwok, J. T. Blockwise adaptivity: Faster training and better generalization in deep learning, 2019. URL https://arxiv.org/abs/1905.09899.

Zheng, T., Deng, Z., Tsang, H. T., Wang, W., Bai, J., Wang, Z., and Song, Y. From automation to autonomy: A survey on large language models in scientific discovery. In Christodoulopoulos, C., Chakraborty, T., Rose, C., and Peng, V. (eds.), *Proceedings of the 2025 Conference on Empirical Methods in Natural Language Processing*, pp. 17733–17750, Suzhou, China, November 2025. Association for Computational Linguistics. ISBN 979-8-89176-332-6. doi: 10.18653/v1/2025.emnlp-main.895. URL https://aclanthology.org/2025.emnlp-main.895/.

# A. Related Works

We survey the landscape of memory-efficient optimization, categorizing prior art into coarse-grained adaptivity, sign-based magnitude reductions, quantized/low-rank approximations, and curvature-aware methods.

## A.1. Coarse-Grained and Layer-Wise Adaptivity

A primary strategy for reducing optimizer memory is to compute adaptive statistics at a granularity coarser than individual elements. **NovoGrad** (Ginsburg et al., 2020) and **BAGM** (Zheng & Kwok, 2019) were among the first to propose layer-wise normalization, computing a single second-moment scalar for each layer. These methods were originally motivated by the observation that different layers in deep networks require distinct learning rates. **LARS** (You et al., 2017) and **LAMB** (You et al., 2019) extended this by scaling updates based on the ratio of weight norms to gradient norms, a technique essential for stabilizing large-batch training in ResNet and BERT architectures.

More recently, this coarse-grained approach has been revisited for LLMs to reduce memory. **AdaLayer** (Zhao et al., 2025) applies block-wise adaptivity but retains full-precision buffers for other statistics. **Adam-mini** (Zhang et al., 2025b) partitions parameters into blocks (based on Hessian structure) and applies a single learning rate per block. Structurally, Adam-mini pulls Adam toward normalized SGDM-like behavior as block sizes become coarse; in the limit where the block size encompasses the entire model, Adam-mini is better described as normalized SGDM. In contrast, BAS explicitly preserves the sign-times-scale update structure of Adam: it keeps coordinate-wise signs while replacing coordinate-wise magnitudes with a shared block-wise magnitude. By maintaining this structure, BAS aligns more closely with Adam's optimization trajectory in our transferability experiments. **AdamS** (Zhang et al., 2025a) simplifies normalization by relying on $(L_0, L_1)$-smoothness assumptions, using the momentum magnitude as a proxy for variance. However, we observed that this formulation struggles to directly transfer AdamW hyperparameters in our pre-training setup, often leading to divergence unless the learning rate is independently retuned. BAS avoids this in the tested regimes by retaining a true variance estimate at block granularity and an Adam-like sign-times-scale update rule, enabling it to serve as a practical AdamW-compatible optimizer.

## A.2. Sign-Based Optimization

A distinct class of memory-efficient optimizers, exemplified by **Signum** (Bernstein et al., 2018) and the more recent **Lion** (Chen et al., 2023), reduces memory overhead by quantizing the update to its sign, effectively using only direction ($\pm 1$) while discarding element-wise magnitude information. While this $\ell_\infty$ geometry reduces the optimizer state footprint significantly, it introduces a fundamental structural mismatch with the adaptive variance scaling ($\ell_2$ geometry) of standard methods like AdamW. By homogenizing the update magnitude across all parameters, sign-based methods lose the ability to perform fine-grained scaling based on parameter-specific sensitivity. Empirical studies have highlighted that this structural divergence makes Lion highly sensitive (Schlotthauer et al., 2025) to hyperparameter selection—requiring distinct tuning recipes (e.g., significantly lower learning rates and higher weight decay) compared to AdamW—and often results in performance degradation on complex reasoning benchmarks where precise convergence is critical. In contrast, BAS is designed to structurally align with AdamW by approximating the second moment, thereby preserving the magnitude-aware scaling essential for stable training dynamics in large-scale language models.

## A.3. Low-Rank Approximations and Quantization

A parallel line of work reduces memory by imposing low-rank structures on the optimizer state. **Adafactor** (Shazeer & Stern, 2018) and **SM3** (Anil et al., 2019) approximate the second-moment matrix using low-rank factorizations (e.g., storing only row and column sums). **CAME** (Luo et al., 2023) improves upon this by introducing a confidence-guided update to reduce instability. Similarly, **GaLore** (Zhao et al., 2024) reduces memory by projecting gradients into a low-rank subspace (e.g., via SVD) before updating, maintaining the optimizer state only for these low-rank components.

Alternatively, numerical precision reduction can be used for compression. **8-bit Optimizers** (Dettmers et al., 2022) utilize block-wise dynamic quantization to store Adam states in INT8 while performing computation in FP32. While effective, this treats quantization primarily as a storage compression technique.

BAS distinguishes itself from these approaches in two ways. First, unlike subspace methods (GaLore), BAS operates on the full parameter space, making it orthogonal to and potentially compatible with low-rank projections. Second, unlike storage-based quantization (8-bit Adam), BAS is *structurally* robust to low precision: because the update depends only

on the sign of the momentum, the first moment can be aggressively quantized (e.g., to FP8) without complex dynamic de-quantization logic.

### A.4. Curvature-Aware and Orthogonal Methods

Standard adaptive methods (Adam) approximate the diagonal of the empirical Fisher information matrix. **Sophia** (Liu et al., 2024) proposes a scalable second-order optimizer that estimates the diagonal Hessian using a lightweight randomized estimator. While Sophia achieves faster convergence in wall-clock time by taking larger steps in flat directions, it relies on element-wise curvature estimates. BAS ignores element-wise curvature in favor of block-wise variance, prioritizing memory efficiency over the faster convergence rate per step offered by second-order information. **Shampoo** (Vyas et al., 2025) and **SOAP** (Vyas et al., 2025) utilize full-matrix preconditioners (approximated via Kronecker products) to better capture parameter correlations. While powerful, these methods typically increase memory and compute overhead. Most recently, **Muon** (Jordan et al., 2024) applies Newton-Schulz iterations to orthogonalize gradients, achieving significant memory savings by discarding the second moment. However, the orthogonalization process introduces substantial communication overhead in distributed settings due to the global reductions required for matrix norms. Furthermore, as noted in Remark 2.2, Muon's dynamics can create a mismatch for Adam-pretrained checkpoints, whereas BAS is designed around Adam-style training recipes.

## B. Detailed Analysis of Optimizer Characteristics

In Table 1, we categorized optimizers based on their structural properties, computational overhead, and memory efficiency. This section provides a granular justification for these classifications, with a specific focus on the arithmetic behind our optimizer state memory claims.

### B.1. Compute Overhead and Update Complexity

**Low Overhead (Element-wise & Block-reduction).** Optimizers classified as "Low" overhead (AdamW, Lion, Adam-mini, AdamS, BAS) utilize update rules that scale linearly with the number of parameters, $\mathcal{O}(d)$.

- **AdamW, Lion and AdamS:** Perform strictly element-wise arithmetic operations.

- **BAS & Adam-mini:** Involve an additional reduction step to compute block-wise statistics (e.g., norms or means). However, as noted in Section 2, calculating the Frobenius norm requires only a single reduction per block. For a parameter tensor of size $4096 \times 4096$ treated as a single block, this adds negligible cost relative to the memory bandwidth required to load the parameters.

**High Overhead (Matrix Factorization).** We classify **Muon** as having "High" compute overhead due to its use of Newton-Schulz iterations. Unlike element-wise updates, Muon requires multiple matrix multiplications ($gg^\top$) per step to orthogonalize the gradients. In distributed training, this introduces two bottlenecks: (1) computational latency from iterative matrix products, and (2) communication latency, as global statistics may be required to maintain orthogonality across sharded states.

### B.2. Hyperparameter Compatibility ("Drop-in" Status)

**Drop-in (AdamW, BAS).** We use "Drop-in" in an empirical sense: an optimizer should transfer AdamW-style hyperparameters (learning rate $\eta$, $\beta$ values) without modification and maintain stable, competitive convergence in the tested regimes. We do not claim this establishes a universal no-retuning guarantee.

- **BAS:** As shown in Figure 4 and Appendix E.1, BAS effectively mimics AdamW's update dynamics in our tested settings. By setting the block-wise scale $\alpha_{\mathbf{i}}^t$ to the aggregate signal-to-noise ratio of the block, BAS produces an update vector with a magnitude comparable to AdamW, allowing AdamW-style learning rate schedules to transfer well.

**Not Drop-in (Lion, Muon).**

- **Lion:** Utilizes a pure sign update scaled only by the learning rate. Lacking the denominator normalization of Adam

($\sqrt{v}$), the effective step size is vastly different, typically requiring $\eta$ to be reduced by $10\times$ and decoupled weight decay to be increased significantly.

- **Muon:** The orthogonalization process fundamentally changes the gradient geometry, making standard Adam-based recipes less directly transferable.

**Partial (Adam-mini, AdamS).** While these methods aim for compatibility, our experiments (Figures 2a and 2b) revealed that applying AdamW hyperparameters directly led to divergence in the volatile early phases of LLM pre-training. Appendix E.1 further separates this transferability behavior from retuned Adam-mini performance.

### B.3. Optimizer State Memory Analysis

A core contribution of BAS is the reduction of optimizer state memory to $\approx 12.5\%$ ($1/8\times$) of AdamW. Here we detail the memory accounting under standard mixed-precision training assumptions.

**Baseline: AdamW (100%)** Standard AdamW maintains two states per parameter: the first moment $m$ and the second moment $v$. To prevent numerical instability (underflow/overflow) during accumulation, these are typically stored in FP32.

- $m$ (FP32): 4 bytes/param

- $v$ (FP32): 4 bytes/param

- **Total: 8 bytes per parameter**.

**Memory-Efficient Baselines: Lion, Adam-mini, Muon, and AdamS($\approx$ 50%)** These methods eliminate the element-wise second moment $v$.

- **Lion/Muon/AdamS:** Maintain only $m$ (FP32). $v$ is discarded.

- **Adam-mini:** Maintains $m$ (FP32). $v$ is stored per-block. Assuming a block size $B \gg 1$ (e.g., 1024 elements), the amortized cost of $v$ is $4/1024 \approx 0$ bytes.

- **Total:** 4 bytes $(m) + \epsilon(v) \approx$ **4 bytes per parameter**.

- **Ratio:** $4/8 = 50\%$.

**Ours: BAS ($\approx$ 12.5%)** BAS achieves a further $4\times$ reduction over the efficient baselines by leveraging the robustness of sign-based updates to quantization.

- **First Moment ($m$):** Unlike SGD-like updates (Adam-mini) or orthogonal updates (Muon) that rely on the precise magnitude of $m$, BAS utilizes only the *sign* of $m$ for the direction (Sign($m$)) and the *norm* of $m$ for the scale. We empirically validated that storing $m$ in **FP8** (1 byte) provides sufficient precision to determine the correct sign and estimate the block-wise norm.

- **Second Moment ($v$):** Stored per-block. With parameter-wise partitioning, this adds one scalar per tensor. For a typical block with $10^7$ parameters, the memory for block scalars is negligible.

- **Total:** 1 byte $(m) + \epsilon(v) \approx$ **1 byte per parameter**.

- **Ratio:** $1/8 = 12.5\%$.

This architectural choice allows BAS to train large models on hardware with significantly less memory capacity, or to fit larger batch sizes on the same hardware, compared to both AdamW and other "memory-efficient" alternatives.

## B.4. Memory Analysis of BAS-MV vs. ZeRO

While BAS-MV offers significant communication advantages, it requires each agent to maintain local optimizer states, preventing the use of ZeRO-style state sharding. In this section, we analyze the total memory footprint (Model + Gradients + Optimizer State) to demonstrate that BAS-MV remains memory-competitive with ZeRO-2 for moderate data-parallel sizes ($N_d$), driven by its highly compressed optimizer state.

We assume a standard mixed-precision training setup where model parameters and gradients are stored in FP16 (2 bytes), while standard AdamW maintains FP32 master weights, momentum, and variance (12 bytes total). We denote the model size (number of parameters) as $\Psi$ and the data-parallel world size as $N_d$. We exclude activation memory, as it is orthogonal to the optimizer choice.

Table 3 details the memory consumption per device for ZeRO-2, ZeRO-3, and BAS-MV.

- **ZeRO-2 (AdamW):** Replicates model parameters ($2\Psi$) but shards gradients and optimizer states across $N_d$ devices. Total memory is dominated by the replicated parameters and the sharded optimizer state fraction.

- **ZeRO-3 (AdamW):** Shards all components (parameters, gradients, and optimizer states), resulting in a footprint that scales linearly with $1/N_d$.

- **BAS-MV:** Replicates parameters and gradients (similar to DDP) but utilizes an extremely compressed local optimizer state ($1\Psi$). As detailed in Section B.3, BAS requires only 1 byte per parameter (FP8 first moment), effectively eliminating the need for separate FP32 master weights due to the robustness of the sign-based update.

*Table 3.* **Per-Device Memory Consumption (Bytes).** Comparison of memory scaling with model size $\Psi$ and data-parallel size $N_d$.

| COMPONENT | ZERO-2 (ADAMW) | ZERO-3 (ADAMW) | **BAS-MV** |
|---|---|---|---|
| PARAMETERS | $2\Psi$ | $\frac{2\Psi}{N_d}$ | $2\Psi$ |
| GRADIENTS | $\frac{2\Psi}{N_d}$ | $\frac{2\Psi}{N_d}$ | $2\Psi$ |
| OPT. STATE | $\frac{12\Psi}{N_d}$ | $\frac{12\Psi}{N_d}$ | **$1\Psi$** |
| **TOTAL** | $\mathbf{2\Psi + \frac{14\Psi}{N_d}}$ | $\mathbf{\frac{16\Psi}{N_d}}$ | $\mathbf{5\Psi}$ |

For typical multi-GPU nodes (e.g., $N_d = 8$), ZeRO-2 requires approximately $2\Psi + 1.75\Psi = 3.75\Psi$ bytes. In comparison, BAS-MV requires a constant $5\Psi$ bytes. The difference ($1.25\Psi$) is relatively minor; for a 1B parameter model, this amounts to roughly 2.5 GB of additional VRAM.

Crucially, for smaller clusters ($N_d = 4$), BAS-MV actually outperforms ZeRO-2 in memory efficiency:

$$\text{ZeRO-2}_{N_d=4} \approx 5.5\Psi \quad > \quad \text{BAS-MV} \approx 5.0\Psi$$

Therefore, BAS-MV offers a memory footprint comparable to ZeRO-2 for moderate data-parallel sizes while providing a substantial reduction in communication overhead. Specifically, by replacing FP16 gradient synchronization (16 bits) with a 1-bit majority vote, **BAS-MV reduces the communication volume by 16×** compared to standard mixed-precision DDP. This method targets bandwidth-constrained environments, particularly those built from commodity GPUs with limited interconnect bandwidth and only a small number of devices. Under these conditions, ZeRO-3 becomes impractical due to its substantial communication overhead, whereas our approach remains efficient.

**BAS-MV Communication Scaling.** For the communication-budget plots, we define the relative communication budget as

$$B_{\text{rel}}(t) = t \cdot C_{\text{alg}}/C_{\text{DDP}},$$

where $C_{\text{alg}}$ is the per-step communication volume of the algorithm and $C_{\text{DDP}}$ is the per-step communication volume of BF16 DDP. BAS and AdamW have $C_{\text{alg}}/C_{\text{DDP}} \approx 1$, while BAS-MV has $C_{\text{alg}}/C_{\text{DDP}} \approx 1/16$ up to negligible block-scalar reductions.

We further evaluated BAS-MV under a fixed 20B-token budget while increasing the data-parallel size from 4 to 128 agents. To reduce endpoint noise, Table 4 reports the mean training loss over the last 50 logged iterations. The loss gradually worsens as DP increases, which is expected because the global batch size is fixed at 256 and the local batch becomes very small at high DP. The main conclusion is stability across this tested DP range, not absence of degradation.

*Table 4.* **BAS-MV fixed-token data-parallel scaling.** Mean training loss over the last 50 logged iterations for Qwen2.5-0.5B trained for 20B tokens.

| DP size | 4 | 8 | 16 | 32 | 64 | 128 |
|---|---|---|---|---|---|---|
| Mean loss | 2.9731 | 2.9998 | 3.0275 | 3.0336 | 3.0433 | 3.0576 |

We also measured wall-clock behavior on 4 NVIDIA GeForce RTX 3090 GPUs connected through PCIe without NVLink. The current BAS-MV implementation is a prototype that packs signs into integer buffers, all-gathers, and unpacks locally, so the result should be interpreted conservatively. Even with this unoptimized implementation, BAS-MV achieved a $7.3\times$ effective per-iteration speedup over DDP in this 4-GPU setting and reached loss 3.1 about $3.09\times$ faster in time-to-target. Table 5 reports additional time-to-target speedups from the same study.

*Table 5.* **BAS-MV time-to-target speedups.** Speedup is measured against BAS/DDP in the 4-GPU PCIe-only RTX 3090 setting.

| Target loss | BAS-MV-4 speedup | BAS-MV-8 speedup |
|---|---|---|
| 3.10 | $1.40\times$ | $3.12\times$ |
| 3.08 | $1.34\times$ | $3.17\times$ |
| 3.06 | $1.26\times$ | $3.05\times$ |
| 3.05 | $1.27\times$ | $3.08\times$ |

These results support BAS-MV for communication-constrained data-parallel LLM training. They do not establish performance in thousand-client federated-learning regimes, which usually have different model sizes, systems assumptions, and local-batch behavior.

## B.5. Block-wise Variance Adaptation and Muon

As discussed in Remark 2.2, while standard Muon lacks variance adaptation, it is theoretically possible to combine the block-wise variance scaling of BAS with the orthogonalization geometry of Muon. We term this Block-wise Adapted Muon (BA-Muon).The algorithm replaces the element-wise sign operation of BAS with the Newton-Schulz iteration (denoted as NewtonSchulz) to approximate the matrix sign function, while retaining the BAS block-wise adaptive scale $\alpha_i^t$ to handle variance.

---

**Algorithm 3** Block-wise Adaptive Muon (BA-Muon)

---

**Require:** $\eta, \lambda, \beta_1, \beta_2, \epsilon$, and partition $x = \{x_\mathbf{1}, \ldots, x_\mathbf{N}\}$
 1: Initialize $m^0 \leftarrow 0$, $v^0 \leftarrow 0$
 2: **while** not converged **do**
 3: $\quad t \leftarrow t + 1$, compute batch gradients $g^t$
 4: $\quad$ **for** each block $\mathbf{i} \in \{1, \ldots, N\}$ **do**
 5: $\qquad$ *// 1. Standard Momentum (First Moment)*
 6: $\qquad m_\mathbf{i}^t \leftarrow \beta_1 m_\mathbf{i}^{t-1} + (1 - \beta_1) g_\mathbf{i}^t$
 7: $\qquad$ *// 2. Block-wise Variance (Second Moment)*
 8: $\qquad v_\mathbf{i}^t \leftarrow \beta_2 v_\mathbf{i}^{t-1} + (1 - \beta_2) \|g_\mathbf{i}^t\|_F^2$
 9: $\qquad$ *// 3. Block-wise Adaptive Scale (Signal-to-Noise Ratio)*
10: $\qquad$ *// Note: Bias correction omitted for brevity*
11: $\qquad \alpha_\mathbf{i}^t \leftarrow \frac{\|m_\mathbf{i}^t\|_F}{\sqrt{v_\mathbf{i}^t} + \epsilon}$
12: $\qquad$ *// 4. Orthogonal Direction (Newton-Schulz)*
13: $\qquad O_\mathbf{i}^t \leftarrow \text{NewtonSchulz}(m_\mathbf{i}^t)$
14: $\qquad$ *// 5. Unified Update*
15: $\qquad$ Let $R_\mathbf{i}, C_\mathbf{i}$ be rows/cols of block $\mathbf{i}$
16: $\qquad x_\mathbf{i}^{t+1} \leftarrow (1 - \eta\lambda)x_\mathbf{i}^t - \eta \cdot \alpha_\mathbf{i}^t \cdot \sqrt{\max(R_\mathbf{i}, C_\mathbf{i})} \odot O_\mathbf{i}^t$
17: $\quad$ **end for**
18: **end while**

---

However, BA-Muon does not outperform Muon in practice. Moonlight (Liu et al., 2025) employs an empirical scaling factor of 0.2, matching the RMS magnitude of AdamW updates. We find that $\alpha_\mathbf{i}^t$ closely matches this value and can substitute for the empirical constant. The underlying reason why such variance adaptation does not substantially enhance Muon's performance, however, warrants further investigation.

**Matrix-Based Optimizer Positioning.** We also added rebuttal comparisons with matrix-based optimizers, including Muon-style methods, in pre-training and SFT settings. Because making these optimizers fully compatible with ZeRO required substantial infrastructure effort, our runs were DDP-only; consequently, wall-clock comparisons against the ZeRO-based BAS/AdamW setup would confound optimizer behavior with implementation differences. The iteration-wise results suggest that matrix-based optimizers can converge faster in pre-training, but their SFT downstream performance was worse than BAS/AdamW in our Adam-pretrained fine-tuning setup. We therefore position BAS more narrowly: it targets memory-constrained Adam-style workflows with low implementation overhead and strong compatibility with AdamW-style recipes, rather than claiming to dominate matrix-based optimizers in every training phase.

# C. Convergence Proof

In the proof, we do not consider weight decay and bias-correction.

**Theorem C.1** (Restatement of Theorem 3.4)**.** *Under Assumptions 3.1-3.3, when $\beta_2 \geq \beta_1$, BAS converges as follows:*

$$
\frac{1}{T} \sum_{t=0}^{T-1} \mathbb{E}\|\nabla f(x^t)\|_1^2 \leq C_{total}\left[ \frac{f(x^0) - f^*}{\eta T} + \frac{Ld}{2}C_\alpha^2 \eta \right.
$$
$$
\left. + \left(2C_\alpha + \frac{C_g}{\epsilon}\right)\left( \frac{C_g}{T(1-\beta_1)} + \frac{dL\eta C_\alpha}{1-\beta_1} + \sqrt{d}\sigma\sqrt{1-\beta_1} \right) \right],
$$

*where $C_\alpha = \sqrt{\frac{1-\beta_1}{1-\beta_2}}$, $C_g$ is an upper bound of $\|\nabla f(x)\|_1$, and $C_{total} = dN\left( \sqrt{C_g^2 + \sigma^2} + \epsilon \right)$.*

*Proof.* By L-smoothness on the update step:

$$f(x^{t+1}) \leq f(x^t) - \eta \sum_{i=1}^{N} \alpha_i^t \langle \nabla_i f(x^t), \text{Sign}(m_i^t) \rangle + \frac{L\eta^2}{2} \sum_{i=1}^{N} \|(\alpha_i^t \cdot \mathbf{1}_{d_i}^\top) \odot \text{Sign}(m_i^t)\|_2^2$$

$$\leq f(x^t) - \eta \sum_{i=1}^{N} \alpha_i^t (\|\nabla_i f(x^t)\|_1 - 2\|m_i^t - \nabla_i f(x^t)\|_1) + \frac{L}{2}\eta^2 \sum_{i=1}^{N} (\alpha_i^t)^2 d_i, \tag{2}$$

where the second inequality is from $\langle a, \text{Sign}(b) \rangle \geq \|a\|_1 - 2\|b - a\|_1$.

Using $\alpha_i^t = \frac{\|m_i^t\|}{\sqrt{v_i^t + \epsilon}}$ and the reverse triangle inequality $\|a\|_2 \geq \|b\|_2 - \|a - b\|_2$:

$$\alpha_i^t \|\nabla_i f(x^t)\|_1 \geq \frac{(\|\nabla_i f(x^t)\|_2 - \|e_i^t\|_2)\|\nabla_i f(x^t)\|_1}{\sqrt{v_i^t} + \epsilon}$$

$$\geq \frac{1}{\sqrt{d_i}} \frac{\|\nabla_i f(x^t)\|_1^2}{\sqrt{v_i^t} + \epsilon} - \frac{\|e_i^t\|_2 \|\nabla_i f(x^t)\|_1}{\sqrt{v_i^t} + \epsilon}, \tag{3}$$

where $e^t := m^t - \nabla f(x^t)$ represents the estimation error between the momentum and the true gradient and the second inequality is from the relation between $l_1$ and $l_2$ norm.

Substituting into (2) and rearranging:

$$\sum_{i=1}^{N} \frac{\eta}{\sqrt{d_i}} \frac{\|\nabla_i f(x^t)\|_1^2}{\sqrt{v_i^t} + \epsilon} \leq f(x^t) - f(x^{t+1}) + 2\eta \sum_{i=1}^{N} \alpha_i^t \|e_i^t\|_1 + \eta \sum_{i=1}^{N} \frac{\|e_i^t\|_2 \|\nabla_i f(x^t)\|_1}{\sqrt{v_i^t} + \epsilon} + \frac{L}{2}\eta^2 \sum_{i=1}^{N} (\alpha_i^t)^2 d_i$$

$$\leq f(x^t) - f(x^{t+1}) + \eta(2C_\alpha + C_g/\epsilon)\|e^t\|_1 + \frac{Ld}{2}\eta^2 C_\alpha^2,$$

where $C_\alpha$ in defined in Lemma C.4 and $C_g$ is an upper bound of $\|\nabla f(x)\|_1$.

Sum from $t = 0$ to $T - 1$ and take the total expectation $\mathbb{E}[\cdot]$.

$$\sum_{t=0}^{T-1} \sum_{i=1}^{N} \frac{\eta}{\sqrt{d_i}} \underbrace{\mathbb{E}\left[\frac{\|\nabla_i f(x^t)\|_1^2}{\sqrt{v_i^t} + \epsilon}\right]}_{\Phi_i^t} \leq f(x^0) - f^* + \eta(2C_\alpha + C_g/\epsilon) \sum_{t=0}^{T-1} \|e^t\|_1 + \frac{Ld}{2}\eta^2 C_\alpha^2 T.$$

We lower bound $\Phi_i^t$ using the Law of Total Expectation, conditioning on $x^t$:

$$\Phi_i^t = \mathbb{E}_{x^t}\left[\mathbb{E}\left[\frac{\|\nabla_i f(x^t)\|_1^2}{\sqrt{v_i^t} + \epsilon}\bigg| x^t\right]\right]$$

$$\geq \mathbb{E}_{x^t}\left[\|\nabla_i f(x^t)\|_1^2 \mathbb{E}\left[\frac{1}{\sqrt{v_i^t} + \epsilon}\bigg| x^t\right]\right]$$

$$\geq \mathbb{E}_{x^t}\left[\frac{\|\nabla_i f(x^t)\|_1^2}{\mathbb{E}[\sqrt{v_i^t} + \epsilon | x^t]}\right]$$

$$\geq \frac{\mathbb{E}[\|\nabla_i f(x^t)\|_1^2]}{\sqrt{C_g^2 + \sigma^2} + \epsilon},$$

where the second inequality by applying Jensen's Inequality to the convex function $h(y) = \frac{1}{\sqrt{y} + \epsilon}$ and the last inequality is from Lemma C.2.

Therefore,

$$\sum_{t=0}^{T-1}\sum_{\mathbf{i}=1}^{N}\mathbb{E}\|\nabla_{\mathbf{i}}f(x^t)\|_1^2 \leq \frac{d\left(\sqrt{C_g^2+\sigma^2}+\epsilon\right)}{\eta}\left[f(x^0)-f^*+\eta(2C_\alpha+C_g/\epsilon)\sum_{t=0}^{T-1}\|e^t\|_1+\frac{Ld}{2}\eta^2C_\alpha^2T\right].$$

We use the norm inequality $\sum_{\mathbf{i}=1}^{N}\|z_\mathbf{i}\|_1^2 \geq \frac{1}{N}\|z\|_1^2$ to recombine the blocks:

$$\frac{1}{T}\sum_{t=0}^{T-1}\mathbb{E}\|\nabla f(x^t)\|_1^2 \leq \frac{N}{T}\sum_{t=0}^{T-1}\sum_{\mathbf{i}=1}^{N}\mathbb{E}\|\nabla_{\mathbf{i}}f(x^t)\|_1^2$$

$$\leq C_{total}\left(\frac{f(x^0)-f^*}{\eta T}+\frac{2C_\alpha+C_g/\epsilon}{T}\sum_{t=0}^{T-1}\mathbb{E}\|e^t\|_1+\frac{Ld}{2}\eta C_\alpha^2\right),$$

where $C_{total} = dN\left(\sqrt{C_g^2+\sigma^2}+\epsilon\right)$.

From Lemma C.3 , the above is

$$\frac{1}{T}\sum_{t=0}^{T-1}\mathbb{E}\|\nabla f(x^t)\|_1^2 \leq C_{total}\left[\frac{f(x^0)-f^*}{\eta T}+\frac{Ld}{2}C_\alpha^2\eta\right.$$

$$\left.+\left(2C_\alpha+\frac{C_g}{\epsilon}\right)\left(\frac{C_g}{T(1-\beta_1)}+\frac{dL\eta C_\alpha}{1-\beta_1}+\sqrt{d}\sigma\sqrt{1-\beta_1}\right)\right].$$

$\square$

**Lemma C.2.** *Under the assumptions in Theorem 3.4, we have*

$$\mathbb{E}[v^t] \leq C_g^2 + \sigma^2.$$

*Proof.* We prove this by induction. When $t=0$, $v^0 = 0 \leq C_g^2 + \sigma^2$, the bound holds trivially.

Assume the bound holds for step $t-1$:
$$\mathbb{E}[v^{t-1}] \leq C_g^2 + \sigma^2.$$

Take total expectation of the update rule for the second moment:

$$\mathbb{E}[v^t] = \beta_2\mathbb{E}[v^{t-1}] + (1-\beta_2)\mathbb{E}[\|g^t\|_2^2]$$
$$= \beta_2\mathbb{E}[v^{t-1}] + (1-\beta_2)(\|\nabla f(x^t)\|_2^2 + \mathbb{E}\|\zeta^t\|_2^2)$$
$$\leq \beta_2(C_g^2+\sigma^2) + (1-\beta_2)(C_g^2+\sigma^2)$$
$$= C_g^2 + \sigma^2.$$

$\square$

**Lemma C.3.** *For the BAS update (ignoring weight decay and bias correction), we have*

$$\frac{1}{T}\sum_{t=0}^{T-1}\mathbb{E}\|e^t\|_1 \leq \frac{C_g}{T(1-\beta_1)}+\frac{dL\eta C_\alpha}{1-\beta_1}+\sqrt{d}\sigma\sqrt{1-\beta_1},$$

*where $e^t := m^t - \nabla f(x^t)$ is the gradient estimation error, $C_g$ is an upper bound of $\|\nabla f(x)\|_1$, and $C_\alpha$ is defined in Lemma C.4.*

*Proof.* Recall the update rule $m^t = \beta_1 m^{t-1} + (1-\beta_1)g^t$. We expand the error $e^t$:

$$e^t = m^t - \nabla f(x^t)$$
$$= \beta_1 m^{t-1} + (1-\beta_1)g^t - \nabla f(x^t).$$

Substitute $m^{t-1} = e^{t-1} + \nabla f(x^{t-1})$:

$$
\begin{aligned}
e^t &= \beta_1(e^{t-1} + \nabla f(x^{t-1})) + (1-\beta_1)g^t - \nabla f(x^t) \\
&= \beta_1 e^{t-1} + \beta_1 \nabla f(x^{t-1}) + (1-\beta_1)(\nabla f(x^t) + \zeta^t) - \nabla f(x^t) \\
&= \beta_1 e^{t-1} + \beta_1[\nabla f(x^{t-1}) - \nabla f(x^t)] + (1-\beta_1)\zeta^t.
\end{aligned}
$$

Let $\Delta^t = \nabla f(x^{t-1}) - \nabla f(x^t)$ be the gradient drift. The recursion is:

$$
e^t = \beta_1 e^{t-1} + \beta_1 \Delta^t + (1-\beta_1)\zeta^t.
$$

Unrolling the recursion from $t$ down to $0$:

$$
e^t = \beta_1^t e^0 + \sum_{k=1}^{t} \beta_1^{t-k+1} \Delta^k + \sum_{k=1}^{t}(1-\beta_1)\beta_1^{t-k}\zeta^k.
$$

Taking expectations and norms (using triangle inequality):

$$
\mathbb{E}[\|e^t\|_1] \leq \underbrace{\beta_1^t \mathbb{E}[\|e^0\|_1]}_{\text{Initial Term } I^t} + \underbrace{\mathbb{E}\left[\left\|\sum_{k=1}^{t}\beta_1^{t-k+1}\Delta^k\right\|_1\right]}_{\text{Drift Term } D^t} + \underbrace{\mathbb{E}\left[\left\|\sum_{k=1}^{t}(1-\beta_1)\beta_1^{t-k}\zeta^k\right\|_1\right]}_{\text{Noise Term } N^t}. \tag{4}
$$

For the initial term sum, we have

$$
\sum_{t=0}^{T-1} I^t = \|\nabla f(x^0)\|_1 \sum_{t=0}^{T-1}\beta_1^t < \frac{C_g}{1-\beta_1}. \tag{5}
$$

Now, we bound the individual drift $\Delta^k$. By $L$-smoothness and the definition of the update step $x^k = x^{k-1} - \eta\alpha^{k-1} \odot \text{Sign}(m^{k-1})$:

$$
\begin{aligned}
\|\Delta^k\|_1 &= \|\nabla f(x^{k-1}) - \nabla f(x^k)\|_1 \\
&\leq \sqrt{d}\|\nabla f(x^{k-1}) - \nabla f(x^k)\|_2 \\
&\leq \sqrt{d}L\|x^{k-1} - x^k\|_2 \\
&= \sqrt{d}L\eta\|\alpha^{k-1} \odot \text{Sign}(m^{k-1})\|_2 \\
&\leq dL\eta C_\alpha
\end{aligned}
$$

Now, substituting this into the drift sum $D^t$:

$$
\begin{aligned}
D^t &\leq \sum_{k=1}^{t}\beta_1^{t-k+1}\mathbb{E}[\|\Delta^k\|_1] \\
&< \frac{dL\eta C_\alpha}{1-\beta_1}
\end{aligned} \tag{6}
$$

Let $Z^t = \sum_{k=1}^{t}(1-\beta_1)\beta_1^{t-k}\zeta^k$. We first bound the expected squared $l_2$-norm

$$
\mathbb{E}[\|Z^t\|_2^2] = \mathbb{E}\left[\left\|\sum_{k=1}^{t}(1-\beta_1)\beta_1^{t-k}\zeta^k\right\|_2^2\right].
$$

Since $\zeta_k$ are independent zero-mean random vectors, the cross-terms $\mathbb{E}[\langle \zeta_i, \zeta_j \rangle]$ are zero for $i \neq j$. Thus, the variance of the sum is the sum of the variances:

$$\mathbb{E}[\|Z^t\|_2^2] = \sum_{k=1}^{t} (1 - \beta_1)^2 \beta_1^{2(t-k)} \mathbb{E}[\|\zeta^k\|_2^2]$$

$$\leq \sigma^2 (1 - \beta_1)^2 \sum_{k=1}^{t} (\beta_1^2)^{t-k}$$

$$= \sigma^2 (1 - \beta_1)^2 \sum_{j=0}^{t-1} (\beta_1^2)^j$$

Using the geometric series inequality $\sum_{j=0}^{t-1} (\beta_1^2)^j < \frac{1}{1-\beta_1^2}$:

$$\mathbb{E}[\|Z^t\|_2^2] < \sigma^2 (1 - \beta_1)^2 \frac{1}{1 - \beta_1^2}$$

$$= \sigma^2 \frac{(1 - \beta_1)^2}{(1 - \beta_1)(1 + \beta_1)}$$

$$= \sigma^2 \frac{1 - \beta_1}{1 + \beta_1}$$

Now, we relate the $l_1$-norm to the $l_2$-norm. By Jensen's inequality and norm equivalence $\|x\|_1 \leq \sqrt{d}\|x\|_2$:

$$N^t = \mathbb{E}[\|Z^t\|_1] \leq \sqrt{d}\mathbb{E}[\|Z^t\|_2]$$

$$\leq \sqrt{d}\sqrt{\mathbb{E}[\|Z^t\|_2^2]}$$

$$< \sqrt{d}\sigma \sqrt{\frac{1 - \beta_1}{1 + \beta_1}}$$

Since $\beta_1 \geq 0$, we have $1 + \beta_1 \geq 1$, so $\sqrt{\frac{1-\beta_1}{1+\beta_1}} \leq \sqrt{1-\beta_1}$.

$$N^t < \sqrt{d}\sigma \sqrt{1 - \beta_1} \tag{7}$$

Substituting (5)-(7) into (4), we arrived at the conclusion.

$\square$

**Lemma C.4** (Upper Bound on Adaptive Scale). *Assume hyperparameters satisfy $0 \leq \beta_1 \leq \beta_2 < 1$. The block adaptive scale $\alpha_{\mathbf{i}}^t$ is uniformly bounded:*

$$\alpha_{\mathbf{i}}^t = \frac{\|m_{\mathbf{i}}^t\|_2}{\sqrt{v_{\mathbf{i}}^t} + \epsilon} \leq \sqrt{\frac{1 - \beta_1}{1 - \beta_2}} \tag{8}$$

*Proof.* To simplify the exposition, we drop the block index $\mathbf{i}$ and establish the result for a single block. In this proof, $\alpha^t \in \mathbb{R}$ denotes the scalar adaptive scale for that block, in contrast to its vector-valued meaning ($\alpha^t \in \mathbb{R}^d$) elsewhere in the paper. Since the argument applies to each block separately, the general multi-block case follows immediately. We expand the moments as sums of past gradients $g^k$ (for $k < t$). Let the EMA weights for the first moment be $w^k = (1 - \beta_1)\beta_1^{t-1-k}$.

$$m^t = \sum_{k=1}^{t-1} w^k g^k \tag{9}$$

We apply the Cauchy-Schwarz inequality to bound the squared Frobenius norm of the sum:

$$\|m^t\|_2^2 = \|\sum_{k=1}^{t-1} \sqrt{w^k}(\sqrt{w^k}g^k)\|_2^2 \tag{10}$$

$$\leq \left(\sum_{k=1}^{t-1} w^k\right)\left(\sum_{k=1}^{t-1} w^k\|g^k\|_2^2\right) \tag{11}$$

Note that the sum of weights forms a geometric series: $\sum_{k=1}^{t-1} w^k = (1-\beta_1)\sum_{j=0}^{t-2}\beta_1^j = 1 - \beta_1^{t-1} < 1$. Therefore:

$$\|m^t\|_2^2 \leq \sum_{k=1}^{t-1}(1-\beta_1)\beta_1^{t-1-k}\|g^k\|_2^2 \tag{12}$$

Now we compare this to the second moment estimator $\nu_t$:

$$v^t = \sum_{k=1}^{t-1}(1-\beta_2)\beta_2^{t-1-k}\|g^k\|_2^2 \tag{13}$$

We examine the ratio of the coefficients for a specific term $\|g^k\|_2^2$. Let $j = t - 1 - k$ be the exponent.

$$\frac{\text{Coeff}_m(k)}{\text{Coeff}_v(k)} = \frac{(1-\beta_1)\beta_1^j}{(1-\beta_2)\beta_2^j} = \frac{1-\beta_1}{1-\beta_2}\left(\frac{\beta_1}{\beta_2}\right)^j \tag{14}$$

Since we assumed $\beta_1 \leq \beta_2$, the ratio $(\frac{\beta_1}{\beta_2})^j \leq 1$ for all $j \geq 0$. Thus, for every $k$, the term in the numerator sum is bounded by $\frac{1-\beta_1}{1-\beta_2}$ times the term in the denominator sum.

$$\|m^t\|_2^2 \leq \frac{1-\beta_1}{1-\beta_2}v^t \tag{15}$$

Taking the square root and dividing by the denominator:

$$\alpha^t = \frac{\|m^t\|_2}{\sqrt{v^t}+\epsilon} \leq \frac{\|m^t\|_2}{\sqrt{v^t}} \leq \sqrt{\frac{1-\beta_1}{1-\beta_2}} \tag{16}$$

$\square$

**Corollary C.5** (Restatement of Corollary 3.5). *Let the stepsize be set as $\eta = \frac{1}{\sqrt{T}}$. Under the assumptions of Theorem 3.4, the algorithm converges to a neighborhood of a stationary point with rate $\mathcal{O}(1/\sqrt{T})$:*

$$\frac{1}{T}\mathbb{E}\sum_{t=0}^{T-1}\|\nabla f(x^t)\|_1^2 \leq \frac{\mathcal{C}_{\text{rate}}}{\sqrt{T}} + \mathcal{C}_{\text{nbd}}$$

*where the convergence constant $\mathcal{C}_{\text{rate}}$ and the neighborhood boundary $\mathcal{C}_{\text{nbd}}$ are defined as (assuming $T \geq 1$):*

$$\mathcal{C}_{\text{rate}} = C_{total}\left[(f^0 - f^*) + \frac{Ld}{2}C_\alpha^2 + \left(2C_\alpha + \frac{C_g}{\epsilon}\right)\frac{C_g + dLC_\alpha}{1-\beta_1}\right],$$

$$\mathcal{C}_{\text{nbd}} = C_{total}\left(2C_\alpha + \frac{C_g}{\epsilon}\right)\sqrt{d}\sigma\sqrt{1-\beta_1},$$

*with $C_{total} = dN\left(\sqrt{C_g^2 + \sigma^2} + \epsilon\right)$.*

*Proof.* Starting from the bound established in Theorem 3.4 and substituting $\eta = T^{-1/2}$. The upper bound becomes:

$$RHS = C_{total} \left[ \frac{f(x^0) - f^*}{T^{1/2}} + \frac{Ld}{2} C_\alpha^2 T^{-1/2} \right.$$
$$\left. + \left( 2C_\alpha + \frac{C_g}{\epsilon} \right) \left( \frac{C_g}{T(1-\beta_1)} + \frac{dLC_\alpha}{T^{1/2}(1-\beta_1)} + \sqrt{d}\sigma\sqrt{1-\beta_1} \right) \right].$$

We organize the terms based on their dependence on $T$:

- **Rate terms ($T^{-1/2}$ and $T^{-1}$):** Noting that for $T \geq 1$, $T^{-1} \leq T^{-1/2}$, the transient terms are dominated by the rate $\mathcal{O}(1/\sqrt{T})$. Collecting these terms yields $\mathcal{C}_{\text{rate}}$.

- **Neighborhood terms ($\mathcal{O}(1)$):** The term independent of $T$ and $\eta$ is the noise floor induced by the stochastic gradients:

$$\mathcal{C}_{\text{nbd}} = C_{total} \left( 2C_\alpha + \frac{C_g}{\epsilon} \right) \sqrt{d}\sigma\sqrt{1-\beta_1}$$

Thus, as $T \to \infty$, the average squared gradient norm converges to $\mathcal{C}_{\text{nbd}}$ at a rate of $\mathcal{O}(1/\sqrt{T})$. $\qquad\square$

# D. Implementation Details

## D.1. Distributed Implementation and Sharding Details

### D.1.1. PARAMETER-WISE PARTITION

Efficient implementation of block-wise adaptivity in modern distributed frameworks (e.g., DeepSpeed ZeRO, PyTorch FSDP) requires careful handling of parameter sharding. In these frameworks, a single logical parameter tensor (e.g., a $4096 \times 4096$ weight matrix) is often physically split into disjoint shards across multiple GPUs.

**The Challenge.** A naive implementation of block-wise adaptivity might attempt to "gather" the full parameter tensor to compute its norm and variance statistics. This would incur a prohibitive communication cost comparable to the gradient exchange itself, negating the efficiency benefits of the optimizer.

**The Efficient BAS Solution.** BAS avoids this overhead by exploiting the additive property of the squared Frobenius norm. For a parameter block $W$ partitioned into $P$ shards $\{W_1, \ldots, W_P\}$ residing on distinct devices, the global squared norm is simply the sum of the local squared norms:

$$\|W\|_F^2 = \sum_{j=1}^P \|W_j\|_F^2 \tag{17}$$

This allows us to compute the global adaptive scale $\alpha_{\mathbf{i}}^t$ using a highly efficient "map-reduce" pattern that avoids gathering the tensor data:

1. **Local Computation:** Each GPU $j$ maintains its local shard of the first moment $m_{\mathbf{i},j}^t$ and computes two local scalars: the squared norm of its moment shard $\|m_{\mathbf{i},j}^t\|_F^2$ and the squared norm of its gradient shard $\|g_{\mathbf{i},j}^t\|_F^2$.

2. **Global Reduction:** An `all_reduce` (SUM) operation is performed on these scalars across all devices. This aggregates the partial sums into global statistics.

3. **Local Update:** Each GPU calculates the global scale $\alpha_{\mathbf{i}}^t$ using the aggregated values and applies the update locally to its shard.

**Communication Cost Analysis.** For a model with $N$ blocks (where $N$ is the number of parameter tensors, typically $10^2$-$10^3$), BAS communicates $2N$ scalars per step. In contrast, synchronizing gradients requires communicating $d$ elements (where $d$ is the number of parameters, typically $10^9$). Since $N \ll d$, the communication overhead for the BAS adaptive scale is negligible. For example, in a 7B model, gradient communication involves gigabytes of data, while BAS scalar reduction involves only kilobytes.

An alternative and highly efficient partitioning strategy is the **shard-wise partition**. In distributed data-parallel training (e.g., using ZeRO or FSDP), large parameters are physically split into "shards" residing on different devices to fit within GPU memory.

**Mechanism.** In this scheme, we treat each *local shard* residing on a device as an independent block **i**. Unlike the parameter-wise partition, which aggregates statistics to recover the global norm of the logical parameter, the shard-wise approach computes the adaptive scale $\alpha_{\mathbf{i}}^t$ solely based on the local statistics of the shard.

**Advantages.** This approach eliminates the need for the scalar all_reduce operations described in Appendix D.1.1, resulting in **zero communication overhead** for the optimizer step. Furthermore, because sharding splits large tensors into smaller, disjoint pieces, this strategy naturally yields a finer block granularity. This moves the algorithm's behavior closer to the element-wise adaptivity of AdamW, which contributes to high stability in practice.

**Limitations and Reproducibility.** The primary drawback of shard-wise partitioning is that the definition of a "block" becomes dependent on the distributed setup. Changing the number of GPUs (world size) or the sharding strategy (e.g., changing ZeRO stages) alters the size and boundaries of the shards, thereby slightly altering the training dynamics. While we observe that BAS is robust to these variations, we adopted the parameter-wise partition as the default for our experiments to ensure strict mathematical reproducibility across different hardware configurations.

## D.2. Additional Experiment Setup

To ensure full reproducibility of our results, we provide detailed specifications regarding experimental setting.

### D.2.1. ALGORITHM IMPLEMENTATION

To ensure reproducibility, we rely on the official implementations of baseline methods (summarized in Table 6) with minor modifications so that they fit in our training framework:

| Method | Official Repository |
|---|---|
| Lion | https://github.com/lucidrains/lion-pytorch |
| AdamS | https://github.com/pku-huzhang/AdamS |
| Adam-mini | https://github.com/zyushun/Adam-mini |

*Table 6.* Official implementation links for baseline methods.

### D.2.2. HYPERPARAMETERS

As highlighted in the main text, our main comparison evaluates whether BAS and other Adam variants can transfer AdamW-style hyperparameters. We therefore apply the optimal AdamW settings to these variants in the transferability experiments. Table 7 lists the exact values used for our main results.

For the Qwen2.5 pre-training runs, we performed a grid search to determine the peak learning rate for both AdamW and Lion. We searched over a logarithmic grid with two values per magnitude (steps of $1 \times 10^{-k}$ and $3 \times 10^{-k}$). We found the optimal peak rates for AdamW to be 3.0e-3 (0.5B) and 1.0e-3 (1.5B), and for Lion to be 1.0e-4 (0.5B) and 3.0e-5 (1.5B). For Supervised Fine-Tuning (SFT) on OLMo, we utilized the AdamW hyperparameters established in the OLMo3 technical report (8.0e-5) and performed a dedicated grid search for Lion using the same granularity, identifying an optimal rate of 3.0e-6. Across all experiments, Lion utilized $\beta_1 = 0.9, \beta_2 = 0.95$.

### D.2.3. FP8 IMPLEMENTATION AND QUANTIZATION

In our FP8 experiments, we utilize a custom packing routine to emulate FP8 formats using standard half-precision (FP16) bitwise representations. We store the first moment buffer ($m_t$) in the e5m2 format (1 sign bit, 5 exponent bits, 2 mantissa bits), which we found superior to e4m3 for representing the dynamic range of gradients.

The quantization process, implemented via bit manipulation on the tensor view, proceeds as follows:

*Table 7.* **Hyperparameter Configuration.** Learning rates (LR) indicate peak values. All runs used a constant stepsize with a 3% warmup period for pre-training and linear decay for SFT. AdamS, Adam-mini, and BAS use the AdamW hyperparameters in the transferability experiments.

| EXPERIMENT | OPTIMIZER | PEAK LR | $\beta_1$ | $\beta_2$ | $\epsilon$ | WEIGHT DECAY |
|---|---|---|---|---|---|---|
| **PRE-TRAINING** | ADAMW | 3.0E-3 | 0.9 | 0.95 | 1E-8 | 0.1 |
| (QWEN 0.5B) | LION | 1.0E-4 | 0.9 | 0.95 | – | 0.1 |
| **PRE-TRAINING** | ADAMW | 1.0E-3 | 0.9 | 0.95 | 1E-8 | 0.1 |
| (QWEN 1.5B) | LION | 3.0E-5 | 0.9 | 0.95 | – | 0.1 |
| **SFT** | ADAMW | 8.0E-5 | 0.9 | 0.95 | 1E-8 | 0.0 |
| (OLMO 7B/32B) | LION | 3.0E-6 | 0.9 | 0.95 | – | 0.0 |

1. **Scaling and Clamping:** To mitigate quantization error and underflow, we first scale the high-precision tensor by a fixed factor $\alpha$ (e.g., $\alpha = 512$). The scaled values are then clamped to the representable range of the target format (e.g., $[-57344, 57344]$ for `e5m2`) to prevent saturation artifacts.

2. **Stochastic Rounding:** unlike standard "round-to-nearest" casting, we implement stochastic rounding to preserve statistical fidelity in the gradients. We interpret the scaled FP16 data as 16-bit integers and inject uniform random noise $\epsilon$ before truncation:

$$t_{\text{int16}} = \text{view\_as\_int16}(t_{\text{fp16}})$$

$$t_{\text{fp8}} = (t_{\text{int16}} + \epsilon) \gg k$$

where $k$ is the number of mantissa bits dropped ($k = 8$ for `e5m2`) and $\epsilon \sim U[0, 2^k - 1]$. This noise injection ensures that small gradients are accumulated probabilistically rather than being zeroed out.

3. **Packing:** The resulting integer is right-shifted by $k$ bits and cast to `uint8` for storage, effectively truncating the lower mantissa bits while preserving the exponent and sign of the original FP16 structure. Unpacking reverses this operation by left-shifting the `uint8` data and dividing by $\alpha$.

- **Casting:** We cast the updated momentum to FP8 immediately before storage and cast back to BF16/FP32 for the update step.

- **Scaling Sensitivity:** Empirical testing confirms convergence is insensitive to larger scaling factors (up to 32,768), provided values remain within the `e5m2` dynamic range.

**Precision Limit of Moment Storage.** We tested more aggressive momentum-state quantization under the same Qwen2.5-0.5B training budget. The 6-bit and 4-bit variants use adaptive per-tensor scaling when repacking the BAS momentum state, rather than the fixed-scale scheme used by the FP8 variant. Table 8 shows a clear threshold: FP8 matches FP32/BF16 in this setup, 6-bit remains trainable but degrades, and 4-bit is unstable under this direct implementation.

*Table 8.* **Precision sensitivity for BAS first-moment storage.** Values are mean losses over the last 50 logged iterations at 16.8B tokens.

| Precision | FP32 | BF16 | FP8 | 6-bit | 4-bit |
|---|---|---|---|---|---|
| Mean loss | 2.8849 | 2.8811 | 2.8847 | 2.9654 | 5.0179 |

Therefore, our supported claim is that FP8 is empirically benign in the tested setting. Sub-8-bit moment storage is a future-work direction and likely requires specialized microscaling, block-wise scaling, or mixed-precision machinery rather than a direct precision reduction.

### D.2.4. BLOCK PARTITIONING SPECIFICS

For the **Parameter-wise** partitioning used in the main experiments, we treated each distinct named parameter in the PyTorch model 'state_dict' as a single block, with two exceptions:

1. **Embeddings** ($W_{emb}$)**:** Due to the sparsity of gradients in vocabulary embeddings, we utilized **row-wise partitioning** (one block per vocabulary token). This prevents updates to frequent tokens from incorrectly scaling the step size for rare tokens.

2. **Output Head** ($W_{head}$)**:** Similarly, the output linear layer was partitioned row-wise to handle the sparse nature of the cross-entropy loss gradient signal.

All other weights (Attention $W_Q, W_K, W_V, W_O$ and MLP up/down projections) used tensor-wise granularity.

## E. Additional Experimental Results

### E.1. Hyperparameter Transferability and Baseline Retuning

This appendix separates two related but distinct questions: tuned optimizer quality and hyperparameter transferability. Because Adam-mini and AdamS are motivated partly by AdamW-style compatibility, our main pre-training results test transferability by applying the AdamW recipe to BAS, Adam-mini, and AdamS. We then add partial retuning studies to clarify whether the observed gaps are solely due to learning-rate mismatch.

**BAS vs. AdamW Transferability.** We performed matched Qwen2.5-0.5B sensitivity studies for BAS and AdamW over learning rate, beta pairs, and weight decay for a 5B-token budget. Across these sweeps, BAS and AdamW tracked each other closely; the last-50-iteration mean loss differences were within approximately 0.02 in the tested settings. This supports the claim that BAS transfers AdamW-style hyperparameters well in this regime. These sweeps are single-seed, so we treat them as evidence for transferability in the tested setup rather than as a universal robustness guarantee.

We also added an SFT learning-rate sweep on Gemma-7B with Alpaca for three epochs. BAS and AdamW again remained close across the tested learning-rate range. Together with the main Qwen2.5 and OLMo3 experiments, this pattern appears across pre-training and SFT, but the appropriate conclusion is that BAS is a promising drop-in candidate for AdamW-style recipes, not that retuning is never useful.

**Adam-mini Transferability and Retuning.** Because BAS is closely related to Adam-mini, we added matched Adam-mini sensitivity experiments on Qwen2.5-0.5B for a 5B-token budget. Table 9 summarizes the learning-rate transferability result. Adam-mini does not match AdamW's optimal learning-rate region in this setup: it improves substantially after lowering the learning rate to $3 \times 10^{-4}$, but remains above AdamW under the same fixed budget.

*Table 9.* **Adam-mini learning-rate sensitivity in the rebuttal study.** Values are last-50-iteration mean training losses for Qwen2.5-0.5B over 5B tokens.

| Optimizer | LR=$3 \times 10^{-4}$ | LR=$1 \times 10^{-3}$ | LR=$3 \times 10^{-3}$ |
|---|---|---|---|
| AdamW | 3.0731 | 3.0547 | 3.0450 |
| Adam-mini | 3.6455 | 17.8783 | 41.0782 |

To separate learning-rate mismatch from the remaining hyperparameters, we additionally swept beta pairs and weight decay at both the transferred AdamW learning rate ($10^{-3}$) and Adam-mini's best tested learning rate ($3 \times 10^{-4}$). At $10^{-3}$, varying beta or weight decay did not bring Adam-mini close to AdamW: the last-50 mean loss ranged from 5.9750 to 70.7010 across beta pairs and from 8.5832 to 36.3928 across weight decays, while AdamW stayed near 3.0. At $3 \times 10^{-4}$, Adam-mini was more stable but remained above AdamW across all tested beta pairs (3.3795–4.2475 vs. 3.0116–3.0761) and weight decays (3.5649–3.6455 vs. 3.0195–3.0547). Thus, in this tested setup, BAS shows stronger AdamW-style transferability than Adam-mini; this is an empirical observation specific to the evaluated regime.

**AdamS and Adam-mini Learning-Rate Sweep.** We additionally conducted learning-rate sweeps for AdamS and Adam-mini on Qwen2.5-0.5B. This original appendix experiment complements the rebuttal retuning study above by showing how both baselines behave after partial learning-rate retuning.

**Observations.** Figure 7 details the validation loss across the learning rate spectrum. We observe the following:

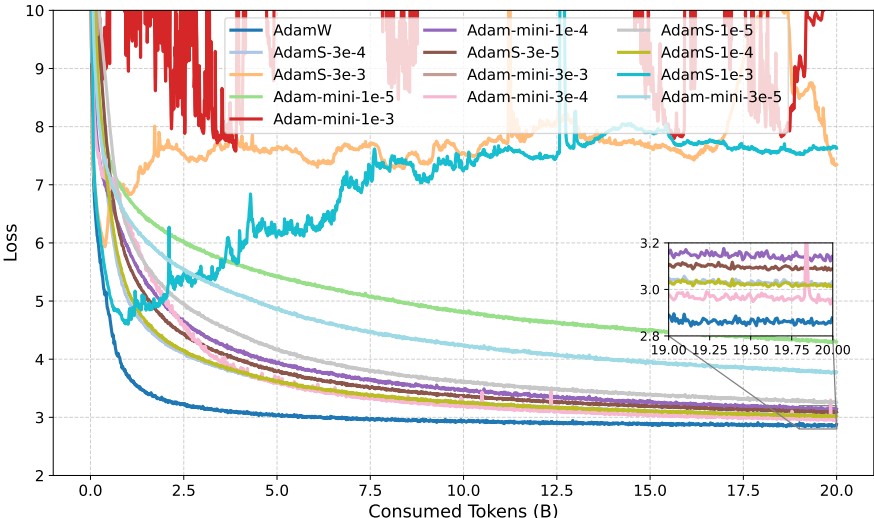

*Figure 7.* **Learning rate sensitivity analysis for AdamS and Adam-mini.** We compare the training loss of AdamW, AdamS, and Adam-mini across a log-scale sweep of learning rates. While AdamS and Adam-mini maintain convergence at lower learning rates (e.g., $\eta \leq 3e-4$), they plateau at a higher loss in this setting. Crucially, the optimal learning-rate regions differ, indicating that the hyperparameter-transferability behavior claimed by AdamS and Adam-mini does not hold in this setting; the optimal $\eta$ for AdamW yields suboptimal or divergent performance for AdamS and Adam-mini.

1. **Convergence vs. Performance:** AdamS and Adam-mini demonstrate an ability to converge at smaller learning rates (e.g., $\eta \leq 3e-4$). However, in this tested setting, the stable configurations plateau at a higher loss compared to AdamW and BAS.

2. **Transferability Gap:** The optimal learning-rate regions differ from AdamW, indicating that AdamS and Adam-mini do not show the same AdamW-style hyperparameter transferability as BAS in this setup.

Consequently, our main results should be interpreted primarily as transferability comparisons, and the learning-rate sweep provides additional context on how the baselines behave after partial retuning.

### E.2. Complete Instruction-Tuning Eval Results

Table 10 presents the comprehensive evaluation results on the Olmo 3 Evaluation Suite. We compare BAS against AdamW, Lion, AdamS, and Adam-mini.

**Performance in the 7B Regime.** In the 7B parameter setting, BAS achieves parity with AdamW across the majority of benchmarks, functioning as an AdamW-compatible optimizer in this tested SFT regime.

- **Robustness across Domains:** BAS matches or slightly exceeds AdamW in critical reasoning and knowledge benchmarks, including **MMLU** (66.2 vs. 66.0), **BigBenchHard** (50.3 vs. 49.5), and **MATH** (59.2 vs. 58.6). This suggests that FP8 first-moment storage in BAS does not impair the model's ability to learn complex representations in this setting.

- **Comparison with Baselines:** In this SFT setup, Lion, AdamS, and Adam-mini lag behind the AdamW baseline on several metrics, often by 1–3 points on metrics like MMLU, HumanEvalPlus, and IFEval. For example, on **AlpacaEval 2 LC**, BAS scores **21.3**, comparable to AdamW's 20.2, while Lion drops to 18.5 and Adam-mini to 18.8.

- **Chat and Tool Use:** BAS demonstrates strong performance in chat and tool-use scenarios, achieving the highest scores in **SimpleQA** (73.4) and **BFCL** (45.2).

**Scalability to the 32B Regime.** To verify that our findings hold at scale, we evaluated BAS on the 32B model. As shown in the rightmost columns of Table 10, BAS remains close to AdamW.

*Table 10.* **Main Results on the Olmo 3 Evaluation Suite.** We compare BAS against 4 baselines in the 7B regime and AdamW in the 32B regime. BAS performance tracks AdamW within the expected variance of the evaluation suite, while structurally different optimizers show consistent degradation.

| | **7B REGIME** | | | | | **32B REGIME** | |
|---|---|---|---|---|---|---|---|
| **BENCHMARK** | ADAMW | LION | ADAMS | ADAM-MINI | **BAS (OURS)** | ADAMW | **BAS (OURS)** |
| ***Math*** | | | | | | | |
| MATH | 58.6 | 56.5 | 57.8 | 57.2 | **59.2** | **69.2** | 69.1 |
| AIME 2024 | 5.2 | 4.1 | **5.4** | 4.2 | 4.9 | 9.0 | **9.2** |
| AIME 2025 | **5.1** | 4.3 | 4.9 | 4.4 | 5.4 | 6.1 | **6.7** |
| OMEGA | **11.1** | 10.1 | 10.4 | 10.2 | 10.8 | **12.7** | 12.5 |
| ***Reasoning*** | | | | | | | |
| BIGBENCHHARD | 49.5 | 47.5 | 48.6 | 48.1 | **50.3** | 66.7 | **67.9** |
| ZEBRALOGIC | **14.5** | 12.9 | 13.5 | 13.7 | 13.7 | 25.4 | **26.7** |
| AGI EVAL | 56.6 | 54.5 | 55.2 | 54.9 | **57.1** | **71.2** | 70.8 |
| ***Coding*** | | | | | | | |
| HUMANEVALPLUS | **65.2** | 60.8 | 62.9 | 63.2 | 63.7 | 74.3 | **75.1** |
| MBPP+ | **51.5** | 46.5 | 49.6 | 49.2 | 50.2 | **56.9** | 56.7 |
| LIVECODEBENCH V3 | **15.9** | 13.9 | 14.5 | 14.2 | 15.7 | 31.0 | **31.9** |
| ***Instruction Following*** | | | | | | | |
| IFEVAL | **76.0** | 72.5 | 73.8 | 73.5 | 75.2 | **85.2** | 84.1 |
| IFBENCH | 25.3 | 24.1 | 24.8 | 25.2 | **26.5** | 27.7 | **28.2** |
| ***Knowledge & QA*** | | | | | | | |
| MMLU | 66.0 | 63.8 | 64.5 | 64.2 | **66.2** | **78.5** | 78.1 |
| POPQA | **15.7** | 14.5 | 14.9 | 14.7 | 15.2 | **23.1** | 21.9 |
| GPQA | 23.9 | 22.5 | 23.2 | 22.9 | **24.8** | **39.9** | 38.0 |
| ***Chat & Tool Use*** | | | | | | | |
| ALPACAEVAL 2 LC | 20.2 | 18.5 | 19.1 | 18.8 | **21.3** | 40.7 | **41.5** |
| SIMPLEQA | 71.0 | 70.5 | 71.2 | 70.8 | **73.4** | **84.2** | 83.5 |
| LITQA2 | **37.1** | 34.8 | 35.5 | 35.1 | 36.8 | **47.1** | 46.5 |
| BFCL | 44.5 | 42.1 | 43.0 | 42.6 | **45.2** | 54.2 | **56.1** |

- BAS achieves slight wins in coding and logic tasks, such as **HumanEvalPlus** (75.1 vs. 74.3) and **ZebraLogic** (26.7 vs. 25.4).

- On knowledge-heavy tasks like **MMLU** and **PopQA**, BAS remains within 0.6–1.2 points of AdamW, which is consistent with the expected variance of large-scale training runs.

hese results support BAS as a practical memory-efficient optimizer that preserves AdamW-level downstream quality in the tested SFT setting, while the broader optimizer-quality comparison should be interpreted alongside the sensitivity studies above.

### E.3. Elaboration on Training Dynamics Metrics

To analyze the heterogeneity of AdamW updates within a parameter block, we examine the effective adaptive scale applied to each element of the parameter tensor. Let $x \in \mathbb{R}^d$ be a flattened parameter block. At step $t$, let $\hat{m}^t$ and $\hat{v}^t$ denote the bias-corrected first and second moments, respectively. The *local adaptive scale* for the $j$-th element, $\tilde{\alpha}_j^t$, is defined as the magnitude of the scaling factor applied to that specific element in AdamW:

$$\tilde{\alpha}_j^t = \frac{|\hat{m}_j^t|}{\sqrt{\hat{v}_j^t + \epsilon}}$$

where $\epsilon$ is a small constant for numerical stability. Let $\alpha_{\mathbf{i}}^t$ denote the scale associated with the $\mathbf{i}$-th block for BAS. With slight abuse of notation, $\mathbf{i}$ denotes the index set of that block. Using these definitions, we track three metrics to quantify the stability and coherence of the layer's training dynamics:

- **Peak Divergence (Max/Mean Ratio):** This metric quantifies the presence of outlier updates within a parameter/block. It is defined as the ratio of the maximum local scale to the mean local scale:

$$[\text{Peak Divergence}]_{\mathbf{i}} = \frac{\max_{j \in \mathbf{i}}(\alpha_j^t)}{\frac{1}{d_{\mathbf{i}}} \sum_{j \in \mathbf{i}} \alpha_j^t} \tag{18}$$

High values indicate that a small subset of parameters (e.g., gated activations in MLPs) are receiving significantly larger updates than the background parameters

- **Scale Dispersion (Fluctuation CV):** This measures the heterogeneity of the update scales within the block. It is calculated as the coefficient of variation (CV) of the local scales:

$$[\text{Scale Dispersion}]_{\mathbf{i}} = \frac{\sqrt{\frac{1}{d_{\mathbf{i}}} \sum_{j=\mathbf{i}} (\tilde{\alpha}_j^t - \text{mean}_{j \in \mathbf{i}}(\tilde{\alpha}_j^t))^2}}{\text{mean}_{j \in \mathbf{i}}(\tilde{\alpha}_j^t)} \tag{19}$$

where $\text{mean}_{j \in \mathbf{i}}(\tilde{\alpha}_j^t)$ is the mean local scale. A higher dispersion indicates non-uniform updates across the layer.

- **Block Coherence (BAS/Mean Ratio):** This metric directly compares the magnitude of the single adaptive scale computed by BAS for a block against the arithmetic mean of the element-wise adaptive scales computed by AdamW. It is defined as:

$$[\text{Block Coherence}]_{\mathbf{i}} = \frac{\alpha_{\mathbf{i}}^t}{\text{mean}_{j \in \mathbf{i}}(\tilde{\alpha}_j^t)} \tag{20}$$

A value near 1.0 indicates that the coarse-grained block-wise scale used by BAS effectively approximates the average fine-grained element-wise scales of AdamW, explaining why BAS mimics AdamW's training trajectory.

