# OpenReview forum: "BAS: Bridging Adam and SignSGD for Memory-Efficient LLM Training"
_ICML.cc/2026/Conference — ICML 2026 regular_

### Official Review · Reviewer_Vdhx · 2026-02-27

**Soundness:** 2
**Presentation:** 2
**Significance:** 2
**Originality:** 2
**Overall Recommendation:** 3
**Confidence:** 4

**Summary:**

This paper presents Block Adaptive Signum (BAS), an optimizer that slashes memory overhead to 12.5% of AdamW by replacing element-wise second moments with block-wise scaling. BAS uniquely preserves Adam’s dynamics, enabling it to inherit existing hyperparameters without retuning, while further optimizing memory via FP8 first-moment storage. Supported by convergence guarantees and a communication-efficient variant (BAS-MV), the method is validated through pre-training and fine-tuning models up to 32B parameters, where it consistently matches AdamW’s performance.

**Compliance With Llm Reviewing Policy:**

Affirmed.

**Final Justification:**

I appreciate the authors’ detailed rebuttal and follow-up clarification. The additional response was helpful in narrowing the intended scope of the claims. In particular, I now better understand that BAS-MV is aimed at reducing communication in the data-parallel regime, that the theoretical contribution is intended as a first guarantee under bounded-gradient assumptions rather than a strongest-possible analysis of Adam-type methods, and that the quantization experiments are meant primarily to justify FP8 as a practical design point rather than to claim robustness to arbitrarily low precision.

That said, my overall evaluation remains unchanged. The paper has clear strengths: it is practically motivated, addresses an important memory bottleneck in AdamW-style optimization, and presents an appealing drop-in design that appears useful in realistic training settings. The empirical results supporting FP8 storage and blockwise second-moment approximation are valuable, and the paper makes a credible case that BAS is practically relevant.

However, the rebuttal and follow-up did not materially change my concerns across soundness, significance, and completeness. On scalability, I appreciate the authors’ clarification that BAS-MV should be judged primarily along the data-parallel axis and that DP=128 is already meaningful for the intended LLM setting. This is a reasonable and helpful clarification. Still, the added evidence remains limited to a relatively small model and a restricted range of settings, with performance degrading gradually as DP increases. As a result, while I agree the method appears stable in the tested regime, I am still not convinced that the paper has established a sufficiently strong case for BAS-MV as a broadly compelling large-scale communication-efficient optimizer variant.

On theory, the follow-up is candid and helpful, but it also confirms that the theoretical limitation remains essentially unchanged. I appreciate the authors’ willingness to narrow the claim and avoid overstating the generality of the analysis. Even so, the current guarantees still rely on restrictive bounded-gradient assumptions, and the gap between the theory and broader realistic optimization regimes remains. This does not invalidate the practical contribution, but it does limit how much weight I can place on the theoretical side of the paper.

**Key Questions For Authors:**

1. Beyond the noted conflicts with ZeRO-style sharding, how does BAS-MV perform in massive-scale distributed environments with hundreds or thousands of devices?
2. The current convergence analysis relies on standard, yet restrictive, assumptions like bounded gradients. Do the authors plan to relax these constraints to better characterize the method’s behavior in non-smooth or heavy-tailed gradient landscapes?
3. Given the success of FP8, it is natural to wonder about the stability limits of the proposed framework. Have you explored (or do you plan to investigate) more aggressive quantization, such as 4-bit, and what are the anticipated trade-offs between precision and optimization stability?

**Limitations:**

While the paper acknowledges the limitations of the BAS-MV variant regarding ZeRO-style sharding, it fails to provide a comprehensive analysis of scaling bottlenecks in other distributed environments. Specifically, the discussion lacks depth regarding the performance of BAS-MV on extremely large models or its efficiency in complex multi-node settings. Furthermore, while the impact statement covers the benefits of accessibility and memory efficiency, it overlooks the potential risks and ethical implications, such as the misuse of increasingly accessible large-scale models.

**Strengths And Weaknesses:**

Strengths:
The paper introduces Block Adaptive Signum (BAS), an optimizer that effectively trades off memory overhead for training performance. By utilizing FP8 storage for the first moment and block-wise approximations for the second, BAS achieves significant memory savings—up to 87.5% relative to AdamW. A particularly strong aspect of the work is its ability to closely mimic AdamW’s dynamics, allowing it to function as a drop-in replacement without requiring hyperparameter retuning. This practicality is supported by solid convergence guarantees and empirical validation on models up to 32B parameters, while the BAS-MV variant further addresses communication efficiency in distributed settings.

Weaknesses:
Despite these contributions, the discussion on limitations remains somewhat narrow. While the authors acknowledge that BAS-MV conflicts with ZeRO-style sharding, they overlook other potential scaling bottlenecks or performance degradation that might emerge in more massive-scale distributed environments. Furthermore, the theoretical analysis relies on simplistic assumptions, such as bounded gradients; relaxing these constraints would better establish the method's generalizability. Finally, while the memory reduction is impressive, the framework’s robustness under more aggressive quantization schemes (e.g., 4-bit) is not investigated, leaving the stability limits of the proposed approach unclear.

---

> ### Author Rebuttal · Authors · 2026-03-31
>
> Thank you for the thoughtful and constructive feedback, and especially for highlighting the practical value of BAS’s memory savings and AdamW compatibility. We appreciate your questions about BAS-MV scaling, the scope of the current theory, and the limits of low-precision moment storage, since these are exactly the areas where the paper benefits from clearer scoping.
>
> All additional numerical results are available at https://anonymous.4open.science/r/BAS_rebuttal-60CF/rebuttal_figure_pack.pdf (hereafter referred to as LINK).
>
> ## Scaling BAS-MV (Question 1)
>
> **We added new Qwen2.5-0.5B BAS-MV runs at larger data-parallel scales, up to 128 devices**  due to time and compute limits, while keeping the total training budget fixed at 20B tokens. This already exceeds the data-parallel scale typically used in the LLM training setting. To reduce endpoint noise, we report the mean training loss over the last 50 logged iterations for each run (see Figure 6 in LINK).
>
> | DP size | 4 | 8 | 16 | 32 | 64 | 128 |
> | --- | ---: | ---: | ---: | ---: | ---: | ---: |
> | Mean loss over last 50 logged iterations | 2.9731 | 2.9998 | 3.0275 | 3.0336 | 3.0433 | 3.0576 |
>
> As the number of devices increases, convergence becomes gradually slower, but we do not observe instability in these runs. Overall, the results support that BAS-MV remains stable across the tested range and is practical for communication-constrained, moderate-scale training.
>
>
> ## On relaxing assumptions (Question 2)
>
> We agree that the bounded-gradient assumption is restrictive from a theoretical standpoint, and we do not present it as matching the strongest recent analyses of Adam-type methods. At the same time, we do not believe it is entirely disconnected from the regime we study. In practical LLM training, weight decay and, in particular, gradient clipping are standard, so a bounded-gradient-style analysis is a reasonable first approximation for the experimental setups considered here. The honest takeaway is therefore that our current guarantees rely on bounded gradients, while broader settings are supported only by empirical stability at present.
>
> A natural next step is to extend the analysis to broader regimes, such as heavy-tailed gradients and non-smooth objectives. We have identified promising prior work in these directions: [1] analyzes convergence of Adam-family methods in non-smooth settings, and [2] provides evidence that sign-based optimizers can remain effective under heavy-tailed noise. These results suggest that some relaxation of our assumptions may be possible. However, doing so would require a substantial reworking of the analysis and is beyond the scope of this paper, whose main emphasis is empirical.
>
>
> ## On the limit of quantization (Question 3)
>
> **We also tested more aggressive low-bit quantization under the same training budget**. As above, we report the mean loss over the last 50 logged iterations (see Figure 7 in LINK). To clarify the implementation, the 6-bit and 4-bit variants use adaptive per-tensor scaling when repacking the SA-Signum momentum state, rather than the fixed-scale scheme used in the FP8 variant. This difference is necessary in practice, since the lower-bit variants tend to diverge much more easily under a fixed scale.
>
> | Precision | FP32 | BF16 | FP8 | 6-bit | 4-bit |
> | --- | ---: | ---: | ---: | ---: | ---: |
> | Mean loss over last 50 logged iterations at 16.8B tokens | 2.8849 | 2.8811 | 2.8847 | 2.9654 | 5.0179 |
>
> These results show a clear threshold under the current implementation. `FP32`, `BF16`, and `FP8` are essentially indistinguishable in this setting, which supports the main-text claim that FP8 is empirically benign. Moving to `6-bit` remains trainable but incurs a clear degradation, while `4-bit` is unstable and far from the baseline loss. So the correct claim is not that BAS is robust to arbitrarily low precision; rather, our current evidence supports FP8, shows noticeable degradation at 6-bit, and treats 4-bit as a limitation of the current implementation.
>
> This also clarifies the practical trade-off: below FP8, we likely need more careful low-bit design rather than a direct drop-in reduction in precision. We will therefore present aggressive low-bit quantization as future work, potentially requiring more advanced schemes such as block-wise or mixed-precision quantization.
>
> ## On Impact Statement
> We will revise the impact statement to explicitly acknowledge potential risks and ethical concerns, including the misuse of more accessible large-scale models, and to emphasize the need for responsible deployment and safeguards.
>
> Please feel free to reach out for additional details or clarification.
>
> [1] Xiao, Nachuan, et al. "Adam-family methods for nonsmooth optimization with convergence guarantees." Journal of Machine Learning Research 25.48 (2024): 1-53.
>
> [2] Yu, Dingzhi, et al. "Sign-Based Optimizers Are Effective Under Heavy-Tailed Noise." arXiv preprint arXiv:2602.07425 (2026).

---

> > ### Author Rebuttal · Reviewer_Vdhx · 2026-04-01
> >
> > Thank you for the detailed rebuttal and the additional experiments. I appreciate that the authors clarified the scope of the current paper more explicitly. However, I do not plan to increase my score, because the rebuttal does not materially change my overall assessment.
> >
> > On large-scale distributed training, the new BAS-MV results are useful, but they remain limited in scope. The added experiments are conducted on a relatively small model (Qwen2.5-0.5B) and only up to 128 devices, and the reported losses degrade gradually as the number of devices increases. While this supports stability in the tested regime, it still does not convincingly address the original concern about behavior in truly massive-scale distributed environments or establish that BAS-MV remains compelling at the scales suggested by the paper’s motivation.
> >
> > On the theoretical side, my concern is largely unchanged. I appreciate the authors’ candid acknowledgement that the bounded-gradient assumption is restrictive, but the rebuttal does not actually relax this assumption or provide stronger guarantees. The argument that gradient clipping makes the analysis more practically relevant is reasonable as motivation, but it does not materially strengthen the theory itself. As a result, the gap between the theoretical treatment and broader realistic optimization regimes remains.
> >
> > The new quantization results are also helpful, but in my view they primarily sharpen the limitation rather than resolve it. The evidence supports FP8 as a practical choice, but also shows clear degradation at 6-bit and instability at 4-bit. This makes the current framework’s robustness margin under more aggressive quantization appear narrower than one might hope.

---

> > > ### Author Response · Authors · 2026-04-02
> > >
> > > We thank the reviewer for the follow-up and for the additional clarification. We clarify below the scope of BAS-MV, the theory claim, and the quantization result.
> > >
> > > ## On Scalability of BAS-MV
> > >
> > > We respectfully disagree that the current BAS-MV experiments are too small-scale to be meaningful. For BAS-MV, the relevant scaling axis is the data-parallel (DP) degree, since BAS-MV is designed to reduce communication in the DP dimension. Under this metric, **DP=128 is already a large regime for modern LLM training**.
> > >
> > > In contemporary large-scale LLM systems, very high total GPU counts are typically achieved through hybrid parallelism, with most of the scaling coming from tensor and pipeline parallelism rather than pure DP. For example, NVIDIA's Megatron configuration for LLaMA-3.1 405B on 1024 GPUs uses DP=8 [1]. Moreover, NVIDIA's Megatron-Bridge documentation explicitly identifies DP >= 128 as a large regime that can already introduce numerical instability [2]. From this perspective, **evaluating BAS-MV up to DP=128 is already sufficient to test it in a practically meaningful large-DP setting for LLM training**.
> > >
> > > **The gradual degradation we observe as DP increases is also expected** under our evaluation protocol, because we keep the global batch size fixed at 256. At DP=128, each GPU receives only 2 samples per step, which is a particularly challenging regime. We therefore view the main result not as "no degradation," but rather as evidence that BAS-MV remains stable even under this extreme small-local-batch setting.
> > >
> > > The reviewer's suggestion of 1000+ devices is more representative of federated learning-style deployments than of mainstream LLM pretraining. Those settings also usually involve **much smaller models** and simpler tasks. For example, FedScale evaluates such scaling with ShuffleNet-V2 and MobileNet-V2, which are about 3M parameters [3], more than 100x smaller than the 0.5B model used here. For this reason, we believe DP=128 already constitutes a strong scalability test for the setting BAS-MV is intended to address. We will revise the paper text to make this scope clearer and avoid suggesting that BAS-MV has already been validated in FL-style thousand-client regimes.
> > >
> > > ## On Theory
> > >
> > > On the theoretical side, we agree that our follow-up neither relaxes the bounded-gradient assumption nor provides a stronger guarantee. Our intended claim is narrower: the paper gives a first convergence guarantee for BAS under bounded-gradient and bounded-variance assumptions, not a theory that matches the strongest recent analyses of Adam-type methods. To avoid overclaiming, **we will revise summary phrases such as "under standard assumptions" to explicitly state this bounded-gradient setting and present broader-regime analysis as future work.**
> > >
> > > ## On Quantization
> > >
> > > **We believe the new quantization results further support our design choice.** In our experiments, FP8 remains stable, whereas 6-bit degrades and 4-bit becomes unstable under the naive quantization setup we used. This behavior is expected: sub-8-bit training typically relies on specialized microscaling or block-wise scaling schemes rather than a plain low-bit format [4,5], and recent 4-bit training results further combine such scaling with additional stabilization techniques [5,6]. **Our current 6-bit and 4-bit tests intentionally do not add this extra machinery, so degradation under a naive implementation is not surprising.** More importantly, the marginal gain below FP8 is much smaller than the main gain already obtained by BAS: relative to AdamW, BAS already saves about 7 bytes per parameter, whereas moving from FP8 to 6-bit or 4-bit would save only an additional 0.25 or 0.5 bytes per parameter, respectively. We therefore view the current evidence as supporting the practical design choice in the paper: FP8 captures most of the memory benefit, while lower precisions require substantially more specialized quantization machinery.
> > >
> > > We hope that the above clarifications help address any remaining concerns from the reviewer.
> > >
> > >
> > > [1] https://docs.nvidia.com/megatron-core/developer-guide/latest/user-guide/training-examples.html
> > >
> > > [2] https://docs.nvidia.com/nemo/megatron-bridge/latest/performance-guide.html
> > >
> > > [3] Lai, Fan, et al. "Fedscale: Benchmarking model and system performance of federated learning at scale." International conference on machine learning. PMLR, 2022.
> > >
> > > [4] Rouhani, Bita Darvish, et al. "Microscaling data formats for deep learning." arXiv preprint arXiv:2310.10537 (2023).
> > >
> > > [5] Abecassis, Felix, et al. "Pretraining large language models with nvfp4." arXiv preprint arXiv:2509.25149 (2025).
> > >
> > > [6] Dettmers, Tim, et al. "Qlora: Efficient finetuning of quantized llms." Advances in neural information processing systems 36 (2023): 10088-10115.

---

### Official Review · Reviewer_ZAG6 · 2026-03-05

**Soundness:** 2
**Presentation:** 1
**Significance:** 2
**Originality:** 2
**Overall Recommendation:** 3
**Confidence:** 2

**Summary:**

This paper proposes an optimizer called BAS that is intended to show Adam-like behavior while using less optimizer memory. The main idea is to replace Adam’s element-wise second-moment adaptivity with block-wise adaptive scaling, while keeping a first-moment update that is similar to sign-based methods. The paper also introduces a communication-efficient distributed variant, BAS-MV, which uses majority-vote sign communication. Empirically, the paper studies both LLM pre-training and fine-tuning, and reports that BAS can match AdamW performance while using significantly less optimizer memory, and its distributed version additionally enables a much lower communication overhead.

**Compliance With Llm Reviewing Policy:**

Affirmed.

**Final Justification:**

The rebuttal was helpful and clarified several points, and I appreciate the additional experiments and effort to narrow some of the original claims.
I am still not fully convinced by the baseline comparisons. In particular, key baselines such as Adam-mini and AdamS are primarily evaluated under transferred AdamW hyperparameters rather than their own tuned settings, which makes these comparisons better interpreted as tests of hyperparameter transferability under an AdamW-style recipe than as fair tuned comparisons of optimizer quality. This makes it difficult to conclude how much of the observed advantage comes from the method itself versus the evaluation protocol. This is especially important given the acknowledged proximity to Adam-mini: despite this, I still do not think the paper provides a sufficiently strong, properly tuned head-to-head comparison against that baseline.
Overall, I think the idea is promising and practically relevant, but the paper still overstates some conclusions relative to the strength of the current empirical evidence.
I remain at a score of 3, but I would be open to revisiting this assessment if other reviewers can provide strong arguments in favor of the paper.

**Key Questions For Authors:**

1. Hyperparameter robustness claim. Can the authors provide a more systematic sensitivity analysis at small scale for BAS vs. AdamW over learning rate, and potentially other hparams? This would help clarify whether the claim holds beyond the specific reported setups.
2. Fairness of the baseline tuning protocol.
In the main experiments, AdamW is tuned, while BAS, Adam-mini, and AdamS are largely evaluated under transferred AdamW settings. Given that Figure 7 shows some non-divergent baseline settings at different learning rates, can the authors clarify how they view the fairness of the comparison, and whether the conclusions in Figure 2a/b still hold under a matched tuning budget for all optimizers?
3. Communication-efficiency metric. Figure 2c is hard to interpret because “Relative Communication Budget” is not clearly defined. Can the authors instead report wall-clock time?

**Limitations:**

The paper includes a limitations section noting that BAS-MV requires local moments and is therefore currently incompatible with ZeRO-style optimizer-state sharding, that the theory assumes bounded gradients and could be relaxed.
It also includes an Impact statement discussing societal impact.

**Strengths And Weaknesses:**

The paper addresses an important problem of reducing optimizer memory and communication overhead while preserving performance at scale. The empirical scope is a strength: the paper studies both pre-training and fine-tuning, including relatively large-scale experiments up to 100B tokens. The high-level intuition behind BAS is also reasonable and easy to follow.

My main concern is that the central empirical claims are stronger than what is actually shown. Specifically, the repeated claims that BAS can directly use the same hyperparameters as Adam, i.e. act as a "drop-in replacement", and work without retuning, are not convincingly supported by the current evaluation. This is a very strong claim and therefore needs substantial evidence.
The results show that BAS works well under the chosen AdamW recipe in the reported settings, but this is not the same as showing robustness to hparams. Supporting such the claim would require a more systematic sensitivity study for BAS and AdamW, at least over learning rate, betas, and weight decay.
This also affects the fairness of the optimizer comparison. The paper states that BAS, Adam-mini, and AdamS are designed to be compatible with AdamW hyperparameters, and therefore only tune AdamW in pre-training and Lion in all settings. However, Figure 2a/b then mainly shows Adam-mini and AdamS failing under that shared setup. This makes it unclear whether BAS is genuinely superior or whether the evaluation protocol is simply unfavorable to the baselines. The appendix learning-rate sweep in Figure 7 already suggests this issue, since some baseline settings no longer diverge there.

I did not find that the ablations were presented clearly. Figure 4 mixes several design choices at once and makes it hard to extract a clear takeaway. This would be much clearer as a table in the main paper. Additionally, Fig. 5 is used to support the claim that BAS has similar update dynamics to AdamW’s, but the figure is dense, and difficult to grasp. The text referring to these figures is also quite sparse.

I was also not convinced by the communication-efficiency presentation. Figure 2c is difficult to interpret because the x-axis, Relative Communication Budget, is underdefined. This would be easier to evaluate if the paper reported wall-clock time. The figure is further weakened by confusing plotting choices, since some overlapping curves are hard to distinguish (blue is used for both adamw and Bas-MV-8)

The paper’s positioning relative to Muon is also confusing. Muon is mentioned several times in the introduction and motivation, and appears again in the appendix through a BAS-Muon discussion, yet it is excluded from the empirical comparison.

Finally, the writing is quite dense. Section 2 would benefit from a clearer definition of the method rather than relying so heavily on the algorithm block.

Overall, I think the core idea is promising and the large-scale experiments are useful, but the paper currently overclaims relative to its evidence and would benefit from a more careful and better-structured empirical study.

---

> ### Author Rebuttal · Authors · 2026-03-31
>
> We thank the reviewer for the thoughtful feedback. We agree that some empirical claims were too broad and that Figures 4/5 and 2(c) were unclear. We will narrow the claims and make the takeaways explicit.
>
> All additional numerical results are available at https://anonymous.4open.science/r/BAS_rebuttal-60CF/rebuttal_figure_pack.pdf (hereafter referred to as LINK).
>
> ## Hyperparameter Sensitivity Analysis (Question 1)
>
> **We conducted a matched sensitivity study on the 0.5B model for 5B tokens**, sweeping learning rate, beta pairs, and weight decay for BAS and AdamW (see Figure 1-3 in LINK).
>
> **Takeaway:** BAS and AdamW show similar sensitivity across sweeps, with last-50 mean loss differences within ~0.02. This supports the claim that BAS matches AdamW’s robustness under the same recipe.
>
> **Limitation:** These are single-seed results. We will state this explicitly and add more seeds if time permits.
>
> ## Fairness of Baseline Tuning (Question 2)
>
> This concern is valid. Figures 2a/b are not fully tuned comparisons, but illustrate hyperparameter transferability: under an AdamW-style recipe, BAS remains stable and close to AdamW, while Adam-mini and AdamS do not. Figure 7 shows these baselines improve with retuning, which we will distinguish from transferability.
>
> At the same time, Figure 7 suggests that, in our setup, even their tuned behavior remains worse than AdamW, while BAS stays close without retuning. **We will narrow the main-text claim accordingly.**
>
> ## Communication Efficiency (Question 3)
>
> We agree that the x-axis in Figure 2(c) is underdefined. It is a communication-budget plot, not a wall-clock or fixed-token plot. We measure cumulative communication in units of one standard BF16-DDP step:
>
> `B_rel(t) = t * (C_alg / C_DDP)`.
>
> Under this definition, AdamW and BAS are about 1, while BAS-MV is about 1/16 because it communicates 1-bit signs instead of 16-bit BF16 gradients; the extra `O(N_blocks)` scalar reductions are negligible since `N_blocks << d`. For the 0.5B run with 19,070 optimization steps, the BAS-MV trajectory ends near 1200.
>
> So Figure 2(c) reflects progress under an equal communication budget. Wall-clock time would be more direct, but our current prototype lacks an optimized 1-bit communication kernel, so raw runtime would confound algorithmic savings with implementation overhead. **We will rename the axis, use clearer colors, and state this limitation explicitly.**
>
> To further prove the effectiveness of BAS-MV, **we scaled the number of devices up to 128** and observed stable training throughout (Figure 6 in LINK).
>
> ## Ablations, Training Dynamics, and Method Clarification
>
> We agree that Figures 4 and 5 are too dense. **We will replace Figure 4 with a compact table and make the takeaways explicit.** The main messages from the 0.5B/20B ablations are: (i) FP8 essentially matches BF16/FP32; (ii) overly coarse partitioning hurts, while shard-wise and parameter-wise partitioning are similar; and (iii) Frobenius norm gives the best efficiency/performance trade-off among the tested norms.
>
> Figure 5 is intended to explain why BAS can share AdamW-like hyperparameters, but the presentation is too dense. **We will rescale the figures accordingly.** The key point is the right panel: BAS does not reproduce AdamW's element-wise scales exactly, but its block scale stays close to AdamW's mean local scale across layers. That is why AdamW hyperparameters transfer reasonably well.
>
> **We will also make the method definition more explicit** in the main text: BAS uses the sign of the first moment for direction and a single Adam-style adaptive scale per parameter block; BAS-MV keeps the same block-wise scale but replaces full gradient synchronization with majority-vote communication of first-moment signs plus negligible scalar reductions. This should make the distinction between BAS, BAS-MV, and other memory-efficient Adam variants clearer.
>
> ## Muon Positioning
>
> Muon is discussed because of its growing popularity. Our position is:
>
> 1. Muon can converge faster per iteration, but requires more infrastructure effort to amortize its overhead.
> 2. Muon follows dynamics that differ more from AdamW, so BAS is a cleaner fit for Adam-style fine-tuning or continued training.
>
> We **added direct Muon/Dion2 [1] comparisons** (Figures 4-5 and Table 1 in LINK). While matrix-based optimizers achieve lower pretraining loss, they yield worse downstream performance in our setup. We therefore position BAS more narrowly: lower implementation overhead and better alignment with memory-constrained Adam-style workflows. **We will revise the paper accordingly**.
>
> The design of BAS-Muon was motivated by the idea of introducing adaptivity into Muon in a simple and elegant way. However, our experiments show that this modification does not yield improved convergence.
>
> We are happy to clarify any points and provide more information if helpful.
>
> [1] Ahn et al. "Dion2: A Simple Method to Shrink Matrix in Muon." arXiv:2512.16928 (2025).

---

> > ### Author Rebuttal · Reviewer_ZAG6 · 2026-04-01
> >
> > Thank you for the clarifications and for the effort in the rebuttal. The additional experiments and explanations were helpful. That said, I still feel some of my concerns remain only partially resolved.
> >
> > The hparam robustness is still not fully convincing to me. I appreciate that you added the sweeps, but the paper’s strongest claims still seem somewhat too strong relative to the evidence. The new results suggest that BAS can behave similarly to AdamW in the tested settings, but I do not think they fully establish the broader “drop-in replacement” or equal-robustness claim. Also, I am still a bit confused by the empirical results relative to AdamW. Since BAS uses sign-based updates together with reduced second-moment information, my prior would be that it should generally behave as an approximation to AdamW and therefore typically be at least slightly worse. Yet in several places, the paper suggests improvements, and I do not think the paper or rebuttal explains in enough detail why this might happen.
> >
> > The rebuttal clarifies that some of the comparisons are intended to demonstrate hyperparameter transferability rather than tuned optimizer quality, which means they are not fully fair comparisons for relative performance. In addition, other reviewers raised the close relationship to Adam-mini, and the rebuttal itself acknowledges Adam-mini as closely related prior work. Given that, I find it strange that there is no properly tuned comparison against Adam-mini.
> >
> > On the systems side, the clarification of the communication metric helps, but the paper still lacks the most direct practical metric of wall-clock time.

---

> > > ### Author Response · Authors · 2026-04-02
> > >
> > > We thank the reviewer for the follow-up. Below we address the remaining questions on hyperparameter sensitivity, Adam-mini, and wall-clock time. We also updated the rebuttal figure-pack link with the new Gemma-7B Alpaca SFT learning-rate sweep and BAS-MV wall-clock figure:
> > >
> > > https://anonymous.4open.science/r/BAS_rebuttal-60CF/rebuttal_figure_pack.pdf
> > >
> > > ## On hyperparameter sensitivity
> > >
> > > In the first-round rebuttal, we added a matched sensitivity study on Qwen2.5-0.5B over learning rate, beta pairs, and weight decay. Across all tested settings, BAS and AdamW stayed very close. This directly addresses the request for a systematic comparison.
> > >
> > > **We now add a second SFT result**: Gemma-7B on Alpaca for 3 epochs under a learning-rate sweep. As shown in Figure 9 of the updated link, BAS and AdamW remain close throughout training.
> > >
> > > Taken together with the paper results, this pattern appears across Qwen2.5-0.5B, Qwen2.5-1.5B, OLMo3-8B, OLMo3-32B, and Gemma-7B; across FineWeb, OLMo-SFT, and Alpaca; and across pre-training and SFT. We therefore make a narrower supported claim: BAS stays close to AdamW across a broad tested range, rather than offering a universal no-retuning guarantee. At the same time, the consistency of this pattern across models, data, and training regimes suggests that BAS is a **promising drop-in candidate** for AdamW in a broader setting.
> > >
> > > ## On the BAS/Adam performance difference
> > >
> > > We suspect part of the confusion is that BAS may look like a fundamentally different sign optimizer. In fact, AdamW itself can be written in the same sign-times-scale form:
> > >
> > > $$
> > > x^{t+1}_j = x^t_j(1-\eta\lambda) - \eta \cdot \frac{|\hat m^t_j|}{\sqrt{\hat v^t_j}+\epsilon}\,\mathrm{Sign}(m^t_j).
> > > $$
> > >
> > > BAS instead uses
> > >
> > > $$
> > > x^{t+1}_j = x^t_j(1-\eta\lambda) - \eta \cdot \alpha^t_{\mathbf{i}}\,\mathrm{Sign}(m^t_j), \qquad j \in \mathbf{i},
> > > $$
> > >
> > > where
> > >
> > > $$\alpha^t_{\mathbf{i}} = \frac{\|\hat m^t_{\mathbf{i}}\|_F}{\sqrt{\hat v^t_{\mathbf{i}}}+\epsilon}$$
> > >
> > > is a shared block-wise scale. Thus, BAS keeps the same first-moment sign structure as AdamW and only replaces AdamW's coordinate-wise adaptive magnitude with a block-wise adaptive magnitude. In the element-wise partition limit, **BAS reduces exactly to AdamW**.
> > >
> > > This also explains why BAS need not always be slightly worse than AdamW. As discussed in the Adam-mini paper [1], **coordinate-wise adaptive scaling is not necessarily optimal** when parameters within a block are strongly correlated; in such cases, a well-chosen block-level scale can match or occasionally outperform fully coordinate-wise scaling. Under this interpretation, it is not surprising that BAS can sometimes perform slightly better and sometimes slightly worse than AdamW.
> > >
> > > At the same time, we do not claim that BAS is uniformly superior to AdamW. Our point is only that occasional improvements over AdamW are plausible under a block-wise geometry, and should not be interpreted as evidence of inconsistency in the comparison.
> > >
> > > ## Fixed budget comparison with Adam-mini
> > >
> > > **A fixed-budget comparison with Adam-mini was provided in Figure 7 of the original manuscript.** Even after a learning rate sweep, Adam-mini converges noticeably more slowly than AdamW under this setting, **including at its optimal learning rate**, and is therefore also slower than BAS.
> > >
> > > Reference [1] states that “Adam-mini is much easier to use … [and] performs well using the same hyperparameters as AdamW,” highlighting hyperparameter transferability as one of its key advantages. In line with this claim, our main-text comparison evaluates Adam-mini under the same hyperparameter settings as AdamW.
> > >
> > > ## Wall-clock time results of BAS-MV
> > >
> > > **We now provide a BAS-MV wall-clock evaluation** on 4 NVIDIA GeForce RTX 3090 GPUs connected through PCIe, without NVLink. We view this as a practically relevant commodity-GPU setting where communication efficiency matters.
> > >
> > > Our current BAS-MV implementation is a prototype rather than a fully optimized communication kernel. Specifically, we pack sign bits into integer buffers, all-gather them, and then unpack them locally. This exposes the communication benefit of the method, but still incurs nontrivial software overhead, so the result should be interpreted conservatively.
> > >
> > > Even under this unoptimized implementation, BAS-MV shows a substantial end-to-end throughput advantage over DDP. In the 4-GPU setup, we observe a **7.3x** effective per-iteration speedup. Combining this with the training curves, BAS-MV reaches loss 3.1 about **3.09x faster in time-to-target** (Figure 10 in the updated link). We therefore view this as evidence that the communication savings can already translate into meaningful gains in a PCIe-only setting, with room for further improvement in more communication-bound regimes or with a more optimized implementation.
> > >
> > > We hope these clarifications address the reviewer's remaining concerns.
> > >
> > >
> > > [1] Zhang et al., "Adam-mini: Use fewer learning rates to gain more," 2024.

---

### Official Review · Reviewer_Tg3Q · 2026-03-12

**Soundness:** 3
**Presentation:** 3
**Significance:** 2
**Originality:** 2
**Overall Recommendation:** 4
**Confidence:** 5

**Summary:**

This paper introduces Block Adaptive Signum (BAS), an optimizer designed to bridge the gap between Adam's adaptivity and the memory efficiency of sign-based methods. BAS partitions model parameters into disjoint blocks and computes a single adaptive scalar per block. This design allows the algorithm to mimic Adam's variance adaptation at a block-wise granularity while reducing second-moment storage to a single scalar per block. By leveraging the inherent robustness of sign-based updates, BAS stores the first moment in FP8 without performance degradation. This strategy reduces the total optimizer state footprint to 12.5% of AdamW's. The authors prove that BAS converges to a neighborhood of a stationary point at a rate of $O(1/\sqrt{T})$. Empirical evaluations on models up to 32B parameters demonstrate that BAS can match AdamW's performance as a drop-in replacemen

**Compliance With Llm Reviewing Policy:**

Affirmed.

**Final Justification:**

I have read the response and will maintain my decision.

**Key Questions For Authors:**

Please see the weaknesses.

**Limitations:**

yes

**Strengths And Weaknesses:**

### Strengths
* **Extreme Memory Efficiency**: By utilizing block-wise second moments and FP8 first-moment storage, BAS reduces optimizer state memory to 12.5% of the standard AdamW requirement.
* **Hyperparameter Compatibility**: BAS structurally aligns with Adam's dynamics closely enough to directly inherit its hyperparameters, allowing it to function as a seamless drop-in replacement in existing pipelines.
* **Communication Efficiency**: The BAS-MV variant leverages sign-based majority voting to reduce communication volume by up to 16x compared to standard distributed data parallel (DDP) training.
* **Large-Scale Validation**: The method is validated through extensive training, including a 1.5B model on 100B tokens and supervised fine-tuning (SFT) of models up to 32B parameters.


---

### Weaknesses

1. **Incremental Novelty Relative to Adam-mini**: While BAS is characterized as a block adaptive Signum with reduced memory overhead, **Adam-mini** has been combine block-wise adaptivity with significant memory savings. BAS appears clearly motivated by the design of Adam-mini, with the primary distinction being the specific use of sign-based updates rather than SGD-like dynamics.

2. **Discrepancy with Established Baseline Results**: The experimental results in this paper report early **divergence for Lion and Adam-mini** during Qwen2.5 pre-training. This does not match results reported in previous works, such as the original papers for Lion and Adam-mini, or thorough comparisons of popular optimizers **[1]**. In those works, the performance of Lion and Adam-mini is very close to Adam without training divergence.

3. **Strong Theoretical Assumptions and Convergence Rates**: The paper proves BAS converges to a neighborhood of a stationary point under the strong **Bounded Gradient assumption** (Assumption 3.3). The authors claim this is a standard outcome for adaptive optimizers; however, this claim is inaccurate. Many previous works, such as **[2-4]**, have proven that Adam can converge to stationary points under much weaker assumptions. Furthermore, even using the $\beta_1, \beta_2$ adjustment techniques from **[2-4]**, it only proves that $\frac{1}{T}\sum _{t=0}^{T-1} \mathbb{E}\|\nabla f(x^t)\|_1^2$ converges to $\mathcal{O}(T^{-1/2})$, which does not match the optimal convergence rate of Adam where $\frac{1}{T}\sum _{t=0}^{T-1} \mathbb{E}\|\nabla f(x^t)\|_2^2 = \mathcal{O}(T^{-1/2})$.

4. **Inaccurate Characterization of Adam-mini’s Collapse**: The statement that "When the entire model is treated as one block, Adam-mini collapses to SGDM, losing the stabilizing effect of sign-based normalization" is inaccurate. Factually, in this case, Adam-mini reduces to **normalized-SGDM**, which has also been shown to be effective for training Transformers.

---

### References
[1] Kaiyue Wen, et al. "Fantastic Pretraining Optimizers and Where to Find Them," arXiv 2509.02046v1.

[2] Haochuan Li, Alexander Rakhlin, and Ali Jadbabaie. "Convergence of Adam under relaxed assumptions." NeurIPS, 2023.

[3] Yusu Hong and Junhong Lin. "On convergence of Adam for stochastic optimization under relaxed assumptions." NeurIPS, 2024.

[4] Peng, Hanyang, et al. "Simple Convergence Proof of Adam From a Sign-like Descent Perspective." arXiv:2507.05966, 2025.

---

> ### Author Rebuttal · Authors · 2026-03-31
>
> Thank you for the detailed feedback and for recognizing BAS's memory efficiency, communication efficiency, and large-scale validation.
>
> All additional numerical results are available at https://anonymous.4open.science/r/BAS_rebuttal-60CF/rebuttal_figure_pack.pdf (hereafter referred to as LINK).
>
> ## Novelty relative to Adam-mini (Weakness 1 & 4)
>
> We agree that Adam-mini is highly relevant prior work and that our earlier wording did not give it enough credit. **BAS is closely related in spirit to Adam-mini and was partly inspired by its key idea** of replacing element-wise second moments with block-wise statistics. We will revise the paper to acknowledge this connection more explicitly. We also agree that our earlier wording about the one-block limit was imprecise: in that setting, Adam-mini reduces to normalized SGDM rather than standard SGDM.
>
> That said, the distinction we intend to emphasize is narrower. Adam-mini still retains coordinate-dependent update magnitudes within a block, whereas BAS uses a shared block-wise magnitude together with coordinate-wise signs. We view this uniform sign-based geometry as the main conceptual difference, and we will revise the paper to present it in this more precise way.
>
> ## Baseline Discrepancy (Weakness 2)
>
> We agree that the mismatch with prior reported results should be addressed explicitly. Figure 2 should not be read as claiming that Lion or Adam-mini are generally unstable. Rather, it reports behavior under our Qwen2.5 pre-training recipe and, crucially, under **transferred AdamW hyperparameters**. Under this setup, **Lion remains stable** but converges somewhat more slowly, while Adam-mini and AdamS become unstable. This can differ from prior works such as [2], which typically tune each optimizer separately. We will revise the text to make this setup dependence explicit.
>
> With that clarification, our empirical claim is narrower than a general statement about optimizer quality. The point of Figure 2 is not that Adam-mini or Lion fail under their own best-tuned recipes, but that Adam-mini and AdamS do not exhibit the same level of **hyperparameter inheritance** as BAS in this setting. This is consistent with Figure 7, where Adam-mini and AdamS become more stable at smaller learning rates but prefer a different learning-rate region from AdamW.
>
> To address the BAS-vs-AdamW part directly, we **added matched 0.5B sensitivity sweeps** (Figures 1-3 in LINK) over learning rate, beta pairs, and weight decay. Across these sweeps, BAS and AdamW track each other closely. We will therefore narrow the paper's wording to: **in the tested setting, BAS remains competitive under the AdamW-style recipe, whereas Adam-mini and AdamS require more retuning and do not show the same transfer behavior.**
>
> ## Theoretical Assumptions (Weakness 3)
>
> We agree that the bounded-gradient assumption is restrictive, and we do not intend to present our theorem as matching the strongest recent analyses of Adam-type methods. Our intended claim is narrower: this paper provides a first convergence result for BAS under bounded-gradient conditions, showing an $O(1/\sqrt{T})$ rate to a neighborhood of a stationary point. We will revise the paper to present the theory in this limited sense rather than under broader wording such as "standard assumptions."
>
> We also agree that several recent works prove stronger guarantees for Adam under weaker assumptions. At the same time, our result is not weaker simply because it is written in terms of the $l_1$ norm: for any gradient vector $g$, $||g||_1 \ge ||g||_2$, so controlling the averaged squared $l_1$ norm is not looser by norm choice alone. However, because the assumptions, proof setting, and convergence target differ, we do not claim that our theorem is stronger than the best existing Adam theory. The fair conclusion is that our theory gives a clean guarantee for BAS, but under a more restrictive assumption set.
>
> We believe the bounded-gradient assumption is reasonable in the context of LLM training. First, standard training typically includes weight decay; combined with bounded sign-based updates, this induces a bounded parameter trajectory. Under mild continuity conditions, this in turn implies bounded gradients along the trajectory. Second, gradient clipping is widely used in practice, which directly enforces bounded gradients during training. Extending the BAS theory to weaker assumptions, such as non-smooth or heavy-tailed settings, is an important direction for future work, but is beyond the scope of the current paper, whose main emphasis is empirical.
>
> We are happy to provide further details and address any additional questions.
>
> [1] Kunstner, F., Yadav, R., Milligan, A., Schmidt, M., & Bietti, A. (2024). Heavy-Tailed Class Imbalance and Why Adam Outperforms Gradient Descent on Language Models. ArXiv, abs/2402.19449.
>
> [2] Kaiyue Wen, et al. "Fantastic Pretraining Optimizers and Where to Find Them," arXiv 2509.02046v1.

---

> > ### Author Rebuttal · Reviewer_Tg3Q · 2026-04-03
> >
> > Thanks for the update. I’ll keep the score unchanged.

---

### Official Review · Reviewer_UxhR · 2026-03-13

**Soundness:** 3
**Presentation:** 3
**Significance:** 3
**Originality:** 3
**Overall Recommendation:** 5
**Confidence:** 3

**Summary:**

This paper presents Block Adaptive Signum (BAS), a memory efficient alternative to AdamW. Instead of storing element-wise second-momentum states, BAS groups parameters into blocks and computes a single adaptive scale per block, while using sign-based updates. This allows shrinking optimizer-state footprint by 8x while keeping Adam's dynamics to directly inherit its parameters.

**Compliance With Llm Reviewing Policy:**

Affirmed.

**Final Justification:**

This paper proposes a simple and well-motivated method. While BAS is very similar to Adam-mini, the fact that it can simply replace AdamW without any HP tuning while requiring 8x less optimizer memory means BAS has high practical importance. Also, I believe that the rebuttal has improved the paper, especially with the new HP transferability experiments affirming BAS's transferrability qualities and showing that Adam-mini does not share this property, which demonstrates the practical benefits of BAS. I recommend authors add these additional experiments in the revision. My other concerns regarding Muon/Dion and BAS-MV were also properly addressed. Therefore, I am raising my score to 5.

**Key Questions For Authors:**

See weaknesses.

**Limitations:**

yes

**Strengths And Weaknesses:**

BAS
### Strengths

1. The proposed method is simple and well-motivated, and the paper is overall well-written.

2. Strong practical optimizer-state memory reduction as BAS requires ~8x less memory compared to AdamW, without requiring HP search.

3. Experiments and ablations are thorough and support the key decisions of the proposed method.


### Weaknesses

1. Regarding Muon, authors mention:

  - Muon introduces computation/communcation overhead in distributed settings: DION [1] proposes a more parallelization-friendly method inspired by Muon and has shown promising results. It would therefore be interesting to compare BAS against Muon and Dion.

  - Authors claim "Muon is unsuitable for fine-tuning or continued training of Adam-pretrained models" (L72): This is an over statement. Liu et al., (2025) shows SFT performance of MUON is on par with Adam on an an Adam-pretrained model, which does not suggest Muon is unsuitable in this case.

2. The communication-efficient variant, BAS-MV, while reducing the communication overhead, may require higher per-device memory than ZeRO-2. More importantly, the experiment evaluating BAS-MV only demonstrates performance in a fixed communication budget, and it is unclear whether this would lead to worse overall performance or not. Authors should consider adding a new experiment with fixed token budget for BAS, BAS-MV, and AdamW, and reporting the resulting performance.

---

> ### Author Rebuttal · Authors · 2026-03-31
>
> Thank you for your constructive feedback and the positive assessment of BAS's memory efficiency and experimental validation.
>
> All **additional numerical results** are available at https://anonymous.4open.science/r/BAS_rebuttal-60CF/rebuttal_figure_pack.pdf (hereafter referred to as LINK).
>
>
> ## Muon/Dion2 Comparison (Weakness 1)
>
> **We conducted experiments comparing Muon/Dion2 against BAS and AdamW in both pre-training (0.5B model, 20B tokens) and SFT (7B model, 3.75B tokens).** See Figure 4-5 and Table 1 in LINK.
>
> Because making matrix-based optimizers fully compatible with ZeRO requires substantial infrastructure effort, our Muon and Dion2 runs are currently DDP-only. As noted in the paper, this is one practical drawback of these methods. A wall-clock comparison would therefore be misleading, so we focus on iteration-wise convergence.
>
> Under this metric, matrix-based optimizers do show faster iteration-wise convergence during pre-training. However, **their SFT downstream task performance is generally worse than BAS/AdamW** (Table 1 in LINK). This is consistent with the pretrain-SFT mismatch suggested by multiple independent sources, as summarized below.
>
> In [1], the authors explicitly stated "A notable phenomenon observed in practice is the suboptimal performance of models pretrained with AdamW when fine-tuned with Muon, and vice versa." in the discussion section. Table 1 of [2] likewise shows Muon underperforming AdamW when SFT is performed on an AdamW-pretrained model. [3] is specifically designed to mitigate this optimizer mismatch when fine-tuning AdamW-pretrained checkpoints. Practitioners have also reported difficulty adopting Muon for RL fine-tuning; for example, [4] attributes this to Muon's more violent updates disrupting delicate "syntax weights."
>
> We agree that “unsuitable” may be too strong. Based on the evidence above, we believe Muon can introduce a noticeable mismatch when fine-tuning models pretrained with Adam. We will soften this claim in the revision to better reflect this nuance.
>
>
>
> ## BAS-MV under Fixed Token Budget and Memory Comparison (Weakness 2)
>
> To further validate BAS-MV, we evaluate it across 4–128 agents under a fixed token budget (20B tokens, 0.5B model). The 4/8-agent results were included in the original submission; **we newly extend to 16–128 agents**. This scale exceeds typical data-parallel regimes in LLM training. To reduce endpoint noise, we report the mean training loss over the last 50 iterations (see Figure 6 in LINK for full results).
>
> | DP size | 4 | 8 | 16 | 32 | 64 | 128 |
> | --- | ---: | ---: | ---: | ---: | ---: | ---: |
> | Mean loss (last 50 iterations) | 2.9731 | 2.9998 | 3.0275 | 3.0336 | 3.0433 | 3.0576 |
>
>
> We observe that BAS-MV exhibits slower convergence on a per-iteration basis, which is expected. However, the degradation is modest and remains stable even at 128 devices. We believe this slight slowdown can be offset by the substantial communication savings (up to 16× compared to DDP).
>
> We do not yet report wall-clock time results, as achieving meaningful gains will require optimized kernels for sign-based communication. Nevertheless, these results suggest that BAS-MV is a promising approach for communication-constrained training scenarios.
>
> In terms of per-device memory, the reviewer is correct that BAS-MV can require more memory than ZeRO-2 at larger DP sizes. Using the Appendix B.4 formulas, ZeRO-2 requires $2\Psi + \frac{14\Psi}{N_d}$, while BAS-MV requires a constant $5\Psi$. This means that at $N_d=8$, ZeRO-2 uses $3.75\Psi$ and BAS-MV uses $5\Psi$, so the gap is only $1.25\Psi$ (about 2.5 GB for a 1B-parameter model). For smaller clusters, BAS-MV can even be more memory efficient: at $N_d=4$, ZeRO-2 uses $5.5\Psi$, compared with $5\Psi$ for BAS-MV. We will revise the text to make this tradeoff explicit: BAS-MV is not universally lower-memory than ZeRO-2, but its overhead is modest at moderate DP sizes and it can be better at small scale, while reducing communication volume by up to 16x.
>
> We welcome any further questions and are glad to elaborate as needed.
>
>
>
> [1] Liu, Jingyuan, et al. "Muon is scalable for llm training." arXiv preprint arXiv:2502.16982 (2025).
>
> [2] Liu, Zehua, et al. "REG: A Regularization Optimizer for Robust Training Dynamics." arXiv preprint arXiv:2510.03691 (2025).
>
> [3] Wang, Zixiao, Yifei Shen, and Huishuai Zhang. "OLion: Approaching the Hadamard Ideal by Intersecting Spectral and $\ell_ {\infty} $ Implicit Biases." arXiv preprint arXiv:2602.01105 (2026).
>
> [4] https://huggingface.co/blog/bird-of-paradise/training-rl-with-muon-1

---

> > ### Author Rebuttal · Reviewer_UxhR · 2026-04-04
> >
> > Thanks to the authors for the detailed rebuttal.
> >
> > I appreciate the added Muon/Dion2 results and discussion. I also found the rebuttal Fig 1-3 very helpful, showing strong transferability of AdamW HP to BAS. In this regard, I believe one important missing experiment is a similar HP transferability study for Adam-mini. While BAS is very similar to Adam-mini, I believe the simplicity of the proposed method and the fact that it inherits AdamW HP while requiring ~8x less memory makes it particularly interesting and useful in practice.
> >
> > While my main concerns regarding BAS-MV and Muon/Dion have been addressed properly, I will wait for the Adam-mini transferability results before revisiting my score.

---

> > > ### Author Response · Authors · 2026-04-04
> > >
> > > We thank the reviewer for the helpful follow-up. We agree that, given the close relation between BAS and Adam-mini, a direct Adam-mini hyperparameter-transferability study is important. We therefore added the requested matched **0.5B / 5B-token Adam-mini study** to the updated figure pack:
> > >
> > > https://anonymous.4open.science/r/BAS_rebuttal-60CF/rebuttal_figure_pack.pdf
> > >
> > > Specifically, we now include **Figures 11-15**, which cover Adam-mini vs AdamW over learning rate, beta pairs, and weight decay. In this tested setting, the learning-rate sweep suggests that Adam-mini does **not** match AdamW's optimal learning-rate region: at `lr=1e-3` and `3e-3`, Adam-mini's last-50 mean losses are **17.8783** and **41.0782**, versus **3.0547** and **3.0450** for AdamW. Adam-mini improves substantially near `lr=3e-4`, although it is still slightly worse than AdamW there (**3.6455** vs **3.0731**).
> > >
> > > To separate the learning-rate mismatch from the remaining hyperparameters, we additionally swept beta pairs and weight decay at both the transferred AdamW learning rate (`lr=1e-3`) and Adam-mini's best tested learning rate (`lr=3e-4`). At `lr=1e-3`, varying beta or weight decay is not sufficient to bring Adam-mini close to AdamW under this fixed budget: the last-50 mean loss ranges from **5.9750** to **70.7010** across beta pairs and from **8.5832** to **36.3928** across weight decays, while AdamW stays near **3.0**. At `lr=3e-4`, Adam-mini becomes much more stable, but still remains above AdamW across all tested beta pairs (**3.3795-4.2475** vs **3.0116-3.0761**) and weight decays (**3.5649-3.6455** vs **3.0195-3.0547**).
> > >
> > > In this tested setting, **BAS appears to exhibit stronger AdamW hyperparameter transferability than Adam-mini**. BAS stays close to AdamW under the AdamW recipe, whereas Adam-mini benefits from learning-rate retuning but still remains somewhat worse than AdamW under the same fixed budget. We emphasize that this is an empirical observation specific to the evaluated setup. We hope these additional Adam-mini sensitivity results address the reviewer's remaining concern.

---

### Decision · Program_Chairs · 2026-04-30

**Decision:**

Accept (regular)

**Comment:**

The proposed method is simple and well-motivated. Experiments show that the proposed method is memory efficient and communication efficient.  Compared with existing methods especially Adam-mini, the idea of the proposed method is not very novel. Furthermore, strong assumptions are needed for convergence proof.